# Targeting USP11 regulation by a novel lithium-organic coordination compound improves neuropathologies and cognitive functions in Alzheimer transgenic mice

Yi Guo [1], Chuanbin Cai[1], Bingjie Zhang[2], Bo Tan[3], Qinmin Tang[1], Zhifeng Lei[1], Xiaolan Qi[1], Jiang Chen [1,4], Xiaojiang Zheng[2], Dan Zi[5], Song Li [6✉] & Jun Tan [1,2,7✉]

## Abstract

**Alzheimer's Disease (AD), as the most common neurodegenerative disease worldwide, severely impairs patients' cognitive functions. Although its exact etiology remains unclear, the abnormal aggregations of misfolded β-amyloid peptide and tau protein are considered pivotal in its pathological progression. Recent studies identify ubiquitin-specific protease 11 (USP11) as the key regulator of tau deubiquitination, exacerbating tau aggregation and AD pathology. Thereby, inhibiting USP11 function, via either blocking USP11 activity or lowering USP11 protein level, may serve as an effective therapeutic strategy against AD. Our research introduces IsoLiPro, a unique lithium isobutyrate-L-proline coordination compound, effectively lowers USP11 protein level and enhances tau ubiquitination in vitro. Additionally, long-term oral administration of IsoLiPro dramatically reduces total and phosphorylated tau levels in AD transgenic mice. Moreover, IsoLiPro also significantly lessens β-amyloid deposition and synaptic damage, improving cognitive functions in these animal models. These results indicate that IsoLiPro, as a novel small-molecule USP11 inhibitor, can effectively alleviate AD-like pathologies and improve cognitive functions, offering promise as a potential multi-targeting therapeutic agent against AD.**

**Keywords** Alzheimer's Disease; Synaptic Damage; Tau; Ubiquitination; USP11 Inhibition
**Subject Category** Neuroscience

## Introduction

Alzheimer's disease (AD), as the most predominant form of age-related dementia, presents a growing challenge to the global aging population. AD is pathologically characterized by senile plaques and neurofibrillary tangles, formed by β-amyloid (Aβ) and hyperphosphorylated tau protein in the brain, respectively, which lead to progressive synaptic and neuronal loss and culminating in cognitive deterioration and dementia (Freitag et al, 2022; Pritam et al, 2022). Currently, over 6.7 million Americans aged over 65 are affected by AD, a figure projected to escalate to 13.8 million by 2060 (2023). The FDA has approved several medications for AD, among which acetylcholinesterase inhibitors (donepezil, galantamine, rivastigmine) and *N*-methyl-D-aspartate receptor antagonist (memantine) primarily offer symptomatic relief without halting or reversing the disease's progression (Kabir et al, 2020). Over the past few decades, drugs for AD development have predominantly focused on targeting Aβ. However, despite numerous targeted therapies being developed, their clinical benefits are generally disappointing (Imbimbo and Watling, 2019), (Karran and De Strooper, 2022). Recently approved monoclonal antibodies targeting Aβ (Aduhelm and Lecanemab) are also under scrutiny because of their limited clinical efficacy and side effects concerns (Karran and De Strooper, 2022; van Dyck et al, 2023).

Recent advances in understanding the pathophysiology of AD have shifted research focus toward targeting pathological tau protein, another crucial factor in AD neuropathology, which is more strongly correlated with synaptic loss and cognitive decline than senile plaques. (Tapia-Rojas et al, 2019) The spectrum of emerging therapeutic strategies includes reducing tau production by antisense oligonucleotides and small interfering RNA, modulating tau's post-translational modifications (PTMs) using small molecule inhibitors, suppressing tau aggregation by various anti-aggregants, enhancing tau degradation with strategies such as proteolysis targeting chimeras, and directly clearing extracellular tau protein by immunotherapies (Congdon et al, 2023). These approaches reflect the increasing recognition of the pivotal roles of tau pathology in AD and the potential for diverse interventions to mitigate its impact.

Various PTMs, including phosphorylation (Ulrich et al, 2018), methylation (Balmik and Chinnathambi, 2021), acetylation (Esteves et al, 2019), and ubiquitination (Flach et al, 2014), are crucial to regulate tau aggregation and clearance. Among all these PTMs, ubiquitination, as one of the most pervasive dynamic

[1]Key Laboratory of Endemic and Ethnic Diseases, Ministry of Education; Key Laboratory of Molecular Biology, Guizhou Medical University, Guiyang 550025 Guizhou, China. [2]Anyu Biotechnology (Hangzhou) Co., Ltd., Hangzhou 310000 Zhejiang, China. [3]Department of Biomedical Sciences, City University of Hong Kong, Hong Kong SAR, China. [4]Department of Pharmacy, School of Medicine, Zhejiang University, Hangzhou 310058 Zhejiang, China. [5]Department of Gynecology, Guizhou Provincial People's Hospital, Guiyang 550025 Guizhou, China. [6]First Affiliated Hospital of Dalian Medical University, Dalian 116021 Liaoning, China. [7]Institute of Translational Medicine; Key Laboratory of Novel Targets and Drug Study for Neural Repair of Zhejiang Province, School of Medicine, Hangzhou City University, Hangzhou 310015 Zhejiang, China.
✉E-mail: lisong@dmu.edu.cn; tanjun@anyuhz.cn

process reversible by deubiquitinating enzymes (DUBs), is central to the quality control of tau protein (Nijman et al, 2005; Wang et al, 2021), via the ubiquitin-proteasome system (UPS) and autophagy-lysosome pathway (ALP) (Ciechanover and Kwon, 2015). Approximately 100 DUBs are encoded in the human genome (Mevissen and Komander, 2017), with the ubiquitin-specific proteases (USPs) representing the largest subgroup (Chen et al, 2021). Currently, several USPs members, such as USP9 (Köglsberger et al, 2017), USP10 (Wei et al, 2022), USP11 (Yan et al, 2022), and USP25 (Zheng et al, 2022), have been identified to accelerate the progression of AD by targeting tau or Aβ precursor protein (APP). Notably, USP11, a DUB located on the X chromosome, has recently garnered attention due to its significant role in women's susceptibility to AD (Yan et al, 2022). USP11 regulates the deubiquitination of tau protein, thereby influencing tau aggregation and exacerbating AD pathology (Yan et al, 2022). This understanding emphasizes the importance of targeting USP11 as a potential therapeutic option for AD.

The development of specific USP11-target inhibitors, either blocking enzyme activity or reducing protein levels, however, is challenged by the high conservation and broad cellular functions of DUBs (Ritorto et al, 2014). The clinical usage of mitoxantrone (MTX), the only one USP11 inhibitor approved by the FDA for anti-cancer therapy, is limited due to serious side effects (Anderson et al, 2022; Lin and Steinmetz, 2018). In our pursuit to overcome these challenges, we have synthesized IsoLiPro, a novel USP11 inhibitor comprising a coordination salt of lithium isobutyrate and L-proline. Impressively, IsoLiPro notably reduced the USP11 protein level and suppressed the USP11-mediated substrate deubiquitination. Rigorous in vitro and in vivo experiments have established that the reduction in USP11 level after IsoLiPro treatment facilitated the ubiquitination and subsequent degradation of tau protein. In addition, chronic oral administration of IsoLiPro in transgenic AD mouse models markedly reduced the levels of both total tau and phosphorylated tau in the brain, thereby alleviating tau pathologies and synaptic loss. Moreover, chronic IsoLiPro treatment also decreased Aβ plaque depositions and significantly improved cognitive performance of AD model mice. Our findings confirm the promise of USP11-targeting therapy against AD and further underscore the potential of IsoLiPro as a novel, orally active small-molecule USP11 inhibitor, offering a promising therapeutic approach for AD.

# Methods

### Reagents and tools table

| Reagent/Resource | Reference or source | Identifier or catalog number |
|---|---|---|
| MG132 | MedChemExpress | Cat:#HY-13259 |
| Cycloheximide | Biotopped | Cat:#C7698 |
| mitoxantrone | MedChemExpress | Cat:#HY-13502 |
| lipofectamine 2000 | Thermo Fisher Scientific | Cat:#116688500 |
| TRIzol reagent | Thermo Fisher Scientific | Cat:#15596026 |
| DAPI-Fluromount-G | SouthernBiotech | Cat:#0100-20 |

| Reagent/Resource | Reference or source | Identifier or catalog number |
|---|---|---|
| Protein A/G agarose beads | Beyotime Biotechnology | Cat:#P2179 |
| **Experimental Models** | | |
| C57BL/6 mice | Shulaibao (Wuhan) Biotechnology Co., Ltd | RRID:IMSR_JAX:000664 |
| 3xTg-AD mice | Shulaibao (Wuhan) Biotechnology Co., Ltd | B6;129-Tg (APPSwe, tauP301L) 1LfaPsen1tm1Mpm/Mmjax, stock No. 004807 |
| 5xFAD mice | Shulaibao (Wuhan) Biotechnology Co., Ltd | B6.Cg-Tg (APPSwFlLon, PSEN1*M146L* L286V) 6799Vas/Mmjax |
| **Recombinant DNA** | | |
| pEGFP-hTau-2N4R | Yunzhou Biosciences (Guangzhou) Co., Ltd | NM_001377265.1 |
| pCMV-HA-Ub | Changsha Zebra Biotechnology Co., Ltd. | NM_021009 |
| pLV-mCherry-USP11 | Changsha Zebra Biotechnology Co., Ltd. | NM_004651.3 |
| pCMV-HA-Ub-K48 | Changsha Zebra Biotechnology Co., Ltd. | NM_021009 |
| **Antibodies** | | |
| Mouse-anti-tau46 | Cell Signaling Technology | Cat:#4019 |
| Mouse-anti-tau12 | Biolegend | Cat:#806501 |
| Mouse-anti-ptau181 | Thermo Fisher Scientific | Cat:#MN1050 |
| Rabbit-anti-ptau202/205 | Abcam | Cat:#ab109930 |
| Rabbit-anti-ptau231 | Abcam | Cat:#ab151559 |
| Rabbit-anti-ptau396 | Abcam | Cat:#ab210703 |
| Mouse-anti-ptau202/205 | Thermo Fisher Scientific | Cat:#MN1020 |
| Mouse-anti-β-Amyloid,17-24 Antibody | Biolegend | Cat:#800703 |
| Rabbit-anti-Iba1 | Abcam | Cat:#ab178846 |
| Mouse-anti-GFAP | Cell Signaling Technology | Cat:#3670 |
| Rabbit-anti-SYN-1 | Abcam | Cat:#20258-1-AP |
| Rabbit-anti-PSD95 | Abcam | Cat:#ab18258 |
| Rabbit-anti-Ubiquitin | Proteintech | Cat:#10201-2-AP |
| Rabbit-anti-Ubiquitin (linkage-specific K48) | Abcam | Cat:#ab140601 |
| Rabbit-anti-Ubiquitin (linkage-specific K63) | Abcam | Cat:#ab179434 |

| Reagent/Resource | Reference or source | Identifier or catalog number |
|---|---|---|
| Rabbit-anti-GAPDH | HUABIO | Cat:#EM1101 |
| Rabbit-anti-USP11 | Abcam | Cat:#ab109232 |
| Rabbit-anti-USP25 | Abcam | Cat:#ab187156 |
| Anti-rabbit IgG | HUABIO | Cat:#HA1001 |
| Anti-mouse IgG | HUABIO | Cat:#HA1006 |
| Anti-mouse IgG, Alexa Fluor™ 488 | Thermo Fisher Scientific | Cat:#A-11001 |
| Anti-rabbit IgG, Alexa Fluor™ 555 | Thermo Fisher Scientific | Cat:#A-21428 |
| **Oligonucleotides and other sequence-based reagents** | | |
| PCR primers | Sangon Biotech | Appendix Tab. S3 |
| **Chemicals, enzymes and other reagents** | | |
| USP11 Recombinant Protein, Human (Sf9) | MedChemExpress | Cat:#HY-P702063 |
| **Software** | | |
| StepOnePlus Real-Time PCR Detection System | AB Applied Biosystems | 272001262 |
| Smart Video Tracking Software 3.0 | Panlab, Harvard Apparatus | |
| **Other** | | |
| Confocal microscope | Olympus | SpinSR10 |
| Rapid GolgiStain Kit | FD neurotechnologies | Cat:#PK401 |
| human $A\beta_{42}$ ELISA Kit | Thermo Fisher Scientific | Cat:#KHB3441 |
| human $A\beta_{40}$ ELISA Kit | Thermo Fisher Scientific | KHB3481 |
| HE staining Kit | Solarbio | Cat:# G1120 |
| Cell Counting Kit-8 assay | Beyotime | Cat:# C0038 |
| DUB Activity Assay Kit | IVDSHOW | Cat:# CN701490 |
| BCA assay kit | Sigma-Aldrich | Cat:#KF016 |
| Reverse transcription reagent kit | Takara | Cat:#RR037 |

## Antibodies, plasmids, and chemicals

Antibodies used in the present study include anti-pTau181 (Thermo Fisher Scientific, MN1050), anti-pTau396 (Abcam, ab109930), anti-pTau231 (Abcam, ab151559), tau46 (Cell Signaling Technology, 4019), anti-pTau202/205 (Abcam, ab210703), AT8 (Thermo Fisher Scientific, MN1020), 4G8 (Biolegend, 800703), anti-Iba1 (Abcam, ab178846), anti-GFAP (Cell Signaling Technology, 3670), anti-SYN-1 (Abcam, 20258-1-AP), anti-PSD95 (Abcam, ab18258), anti-Ubiquitin (Proteintech, 10201-2-AP), anti-Ubiquitin (linkage-specific K48, Abcam, ab140601), anti-Ubiquitin (linkage-specific K63, Abcam,

ab179434), anti-GAPDH (HUABIO, EM1101), tau12 (Biolegend, 806501), USP11 (Abcam, ab109232), and USP25 (Abcam, ab187156). Plasmids, including pEGFP-hTau-2N4R, pCMV-HA-Ub, and pLV-mCherry-USP11 were sequenced and prepared using an endotoxin-free plasmid extraction kit (Tiangen). Proteasome inhibitor MG132 (MedChemExpress, HY-13259), protein synthesis inhibitor cycloheximide (CHX, biotopped, C7698), and USP11 inhibitor mitoxantrone (MedChemExpress, HY-13502) were purchased commercially.

## Cell culture, transfection, and treatments

SH-SY5Y human neuroblastoma cells or SH-SY5Y human neuroblastoma cells stably expressing wild-type (WT) full-length human tau (SH-SY5Y-hTau) were cultured in DME/F-12 medium (Gibco) containing 10% fetal bovine serum (FBS) (Biological Industries, 04-001-1ACS) and penicillin (100 U/mL)/streptomycin (100 μg/mL) in a 37 °C incubator with humidified atmosphere of 5% $CO_2$. As for the construction of human embryonic kidney 293 cells (HEK293) overexpressing WT full-length human tau (HEK293-hTau) and HEK293 cells overexpressing both human tau and USP11 (HEK293-hTau/USP11), HEK293 cells were cultured in DMEM/High Glucose medium (Gibco) containing 10% FBS in a humidified atmosphere of 5% $CO_2$ at 37 °C. DNA plasmids were transiently transfected in HEK293 cells using lipofectamine 2000 (Invitrogen, 116688500). Four to six hours post-transfection, the medium was replaced with a new complete medium. Tau441 lentivirus were used to transduce SH-SY5Y and induced pluripotent stem cells (iPSCs). To inhibit proteasome activity, SH-SY5Y-hTau cells were cultured with DME/F-12 medium containing 25 mM IsoLiPro with or without proteasome inhibitor MG132 (10 μM), or the same volume of vehicle (DMSO) for 24 h (h) at 37 °C.

## Animals and treatments

Female C57BL/6 mice (2-month and 9.5-month-old) were purchased from Shulaibao (Wuhan) Biotechnology Co., Ltd. The 9.5-month-old 3xTg-AD mice (B6;129-Tg (APPSwe, tauP301L) 1LfaPsen1$^{tm1Mpm}$/Mmjax, stock No. 004807) carrying three human transgenes (APPswe, PS1M146V, and tauP301L) and the 2-month-old 5xFAD mice (B6.Cg-Tg (APPSwFlLon, PSEN1*M146L* L286V) 6799Vas/Mmjax) carrying three human transgenes (APPswe, APPflo, APPlon, PS1M146V, PS1L286V) were pursed from Shulaibao (Wuhan) Biotechnology Co., Ltd. The 3xTg-AD mice, 5xFAD mice, and their age-matched WT littermates were administered IsoLiPro (560 mg/kg/day) or an equivalent amount of sterile water by gavage for continuous 12 and 16 weeks, respectively. All mice were housed under standard conditions (24 ± 2 °C room temperature, 12 h light/dark cycle) with free access to food and water.

## Powder X-ray diffraction (PXRD)

Powder X-ray diffraction data were collected using a Bruker D8 ADVANCE X-ray diffractometer (Bruker, Germany). The experimental conditions were as follows: operating voltage of 40 kV, operating current of 40 mA, step time of 0.3 s, step size of 0.02° (2θ), and data collection range from 3° to 40° (2θ).

## $^1$H nuclear magnetic resonance ($^1$H NMR)

$^1$H NMR data were collected at room temperature using an Avance III 400 MHz NMR spectrometer (Bruker, Germany), with CD3OD as the deuterated solvent.

## Fourier transform infrared analysis (FT-IR)

FT-IR spectra were acquired at room temperature using a Frontier Mid-IR FT-IR spectrometer (Perkin Elmer, USA). The samples were ground uniformly with dried potassium bromide, and then pressed into transparent pellets under 1.0 ton pressure for 2 min using a pellet press. The infrared detection range was from 4000 to 500 cm$^{-1}$, with a resolution of 0.2 cm$^{-1}$.

## Differential scanning calorimetry (DSC)

DSC data were collected using a Discovery DSC 250 differential scanning calorimeter (TA, Instruments, USA). Accurately weighed samples were placed in perforated DSC sample pans, and the exact mass of each sample was recorded. The samples were then heated at a rate of 10 °C/min to the final temperature.

## Magnetic resonance mass spectrometry (MRMS)

MRMS data were collected using a solariX MRMS instrument (Bruker, Germany). The mass spectrometry ion source was electrospray ionization (ESI), with a mobile phase consisting of methanol and water (1:1, v/v), operated in negative ion mode. The scan range was from 57.7 to 1000 m/z.

## Single crystal X-ray diffraction (SC-XRD)

SC-XRD data were collected using a Bruker D8 Venture X-ray diffractometer equipped with a graphite monochromator and Mo-Kα radiation (λ = 0.71073 Å). A crystal (0.18 mm × 0.09 mm × 0.02 mm) was analyzed at 170 K using φ and ω scans. The crystal structure was solved with ShelXT software via direct methods, and non-hydrogen atoms were located through different Fourier synthesis. Anisotropic temperature factors for non-hydrogen atoms were refined using full-matrix least-squares methods. Hydrogen atoms were positioned by difference Fourier synthesis and refined using the riding mode.

## Molecular docking

Molecular docking between compounds and USP11 was performed using Autodock Vina. The crystal structure of USP11 (PDB ID: 4MEL) was obtained from the RCSB protein data bank (http://www.rcsb.org). The lowest-energy docking solutions from the top 100 search results were chosen. The docking results were visualized and analyzed by pymol software.

## Molecular dynamics (MD) simulation

Based on the molecular docking results, molecular dynamics simulations were performed using Amber 18 to investigate the critical interactions between compounds and the USP11 protein. The initial topology file was obtained using the LEAP module, and amino acid detection was performed using the Amber ff14SB force field. TIP3P water model was

selected, and Na$^+$ was added to make the system electrically neutral. The box was set to a truncation value of 10 Å. The steepest descent method and the conjugate gradient method were used to minimize the system. The system was heated in a canonical ensemble (NVT) and balanced in an isothermal-isobaric ensemble (NPT), and finally, the temperature of the whole system was maintained at 300 K by the Langevin thermostat for 100 ns dynamics simulation. Basic parameters of trajectory analyses such as root mean square deviations (RMSD), root mean square fluctuation (RMSF), the radius of gyration (RG), number of hydrogen bonds (H-bonds) were analyzed for each protein–ligand complex. The MM/PBSA method was used to calculate the binding free energy (ΔG) after the system stability.

## Microscale thermophoresis (MST)

The binding affinity between recombinant USP11 protein and IsoLiPro was estimated using a microscale thermophoresis assay. Recombinant USP11 protein was labeled with RED-tris-NTA protein labeling kit (NanoTemper, MO-L018) according to standard protocol. The working samples were prepared by 1:1 serial dilution of IsoLiPro, up to 16 times, in MST optimized buffer (50 mM HEPES, pH 7.5, 500 mM NaCl, 5% Glycerol, 1 mM TCEP). Then, a constant concentration of 40 nM labeled recombinant USP11 protein was introduced into each sample. Equal volumes (10 μL) of binding reactions were mixed by pipetting and incubated for 5 min at room temperature. Mixtures were enclosed in standard-treated or premium-coated glass capillaries and loaded into the instrument (Monolith NT.115, NanoTemper, Germany).

## Western blotting

The cells or brain tissues were homogenized for 30 min in buffer (pH 7.6, 50 mM Tris-HCl, 10 mM dithiothreitol, 2% sodium dodecyl sulfate, 10% glycerol, and 0.2% bromophenol blue) containing protease inhibitors cocktail on ice, and then centrifuged at 12,000 × g at 4 °C for 20 min. The supernatant was collected, and the protein levels were determined using a bicinchoninic acid (BCA) assay kit (KF016, Sigma-Aldrich) by following the manufacturer's instructions (served as soluble protein). The pellet was further incubated with 8% (wt/vol) SDS buffer at 4 °C and then ultrasonicated to reach a complete resuspension. The suspension was centrifuged (18,000 × g, 4 °C for 20 min), and the supernatant was collected (served as an insoluble fraction). The supernatants of different factions were then respectively mixed with loading buffer (3:1, vol/vol) containing 200 mM Tris-HCl, pH 6.8, 8% SDS, 40% glycerol, and boiled for 10 min. For Western blotting, the proteins were separated by 10% SDS-polyacrylamide gel via electrophoresis for about 2 h and transferred to nitrocellulose membranes (0.45 nm) for 1 h. The membranes were blocked with 5% (wt/vol) BSA dissolved in PBS for 1 h and incubated with primary antibodies overnight at 4 °C. Membranes were then incubated with a secondary antibody for 1 h at room temperature and visualized using the ECL Imaging System (GeneGnome XRQ). Immunoreactive bands were quantitatively analyzed by Image J software.

## Immunohistochemistry staining analysis

For immunohistochemistry (IHC) staining, the slides were incubated at 70 °C for 40 min and then deparaffinized. Next, we continued to

retrieve the antigen using citrate buffer and blocking endogenous peroxidase with 3% H$_2$O$_2$. Nonspecific binding sites were blocked with 5% BSA for 30 min at 37 °C. Then sections were incubated with primary antibodies overnight at 4 °C, followed by incubation with HRP-labeled secondary antibodies for 1 h at 37 °C. The reaction was visualized using DAB, and the slides were counterstained with hematoxylin. Finally, they were sequentially dehydrated in 50, 75, 85, 95, and 100% ethanol five times, cleared in xylene two times (15 min each), and coverslipped with Permount solution.

## Quantitative real-time PCR

Total RNA was extracted using TRIzol reagent (Thermo Fisher Scientific, 15596026), and 1 μg of total RNA was reverse-transcribed into complementary DNA (cDNA) using the reverse transcription reagent kit (RR037, Takara). Real-time PCR was performed using a StepOnePlus Real-Time PCR Detection System (AB Applied Biosystems, 272001262, Cossell Biotechnology). The PCR system contains 2 μL forward and reverse primers, 10 μL SYBR Green PCR master mixes, 1 μL cDNA, and 7 μL diethylpyrocarbonate (DEPC H$_2$O). The primers used in this study were listed as follows: *USP11*-forward: GTTCCACTGCCTATCAGCCACAAG, *USP11*-reverse: GAAGACATCAGCCACCATCATCCTC; *tau*-forward: AGAACGCCAAAGCCAAGACAGA, *tau*-reverse: CATTGCTGAGATGCCGTGGAGAG; *GAPDH*-forward: TCAAGGCTGAGAACGGGAAG; *GAPDH*-reverse: CGCCCCACTTGATTTTGGAG. The $2^{-\Delta\Delta Ct}$ method was used to calculate relative gene expression after normalization to the *GAPDH* internal control.

## In vitro USP11 activity assay

The C-terminal conjugate of ubiquitin with 7-amino-4-methylcoumarin (Ub-AMC) was used to measure the deubiquitinase activity of USP11 (IVDSHOW, CN701490). The fluorescence intensity was detected using a microplate reader with a 360 nm excitation/460 nm emission optic module for 30 min. All experiments were conducted in triplicate, and the mean values were analyzed.

## Immunoprecipitation (IP) assay

The total proteins were extracted by lysis buffer containing protease inhibitors and measured the concentration using a BCA assay kit (KF016, Sigma-Aldrich). Equal amounts of proteins were incubated with primary antibody at 4 °C overnight on a shaking platform and then mixed with 20 μL of pretreated protein A/G agarose beads (P2179, Beyotime Biotechnology). After being mixed, the samples were incubated at 4 °C on a rocking platform for 1 h, followed by magnetic separation and supernatant discarding. The immune complex was washed using precooled lysate (without protease inhibitor) to remove unbound proteins five times. 1 × loading buffer was added to the samples and heated at 100 °C for 10 min to elute the immune complex from the magnetic beads. Finally, the samples were subjected to Western blotting.

## Immunofluorescence staining analysis

For immunofluorescent (IF) staining, brain sections and cultured cells were fixed in 4% (vol/vol) paraformaldehyde for 30 min at room temperature and permeabilized in 0.5% Triton x-100

(vol/vol) diluted in PBS. Nonspecific binding sites were blocked via incubation in 5% (wt/vol) BSA containing 0.1% Triton x-100 (vol/vol) for 1 h. The samples were incubated with primary antibodies at 4 °C overnight, followed by washing three times in PBS and subsequent incubation with Alexa Fluor 488– and Alexa Fluor 594–conjugated secondary antibodies (Thermo Fisher Scientific, A11001, A11005) for 1 h at room temperature. After washing three times in PBS, DAPI-Fluoromount-G was used for nuclear DNA staining (SBA, 0100-20). Images were observed and captured by a confocal microscope (SpinSR10, Olympus).

## Cycloheximide protein turnover assay

SH-SY5Y-hTau cells were treated with DMSO or 100 μg/mL CHX for the indicated time points with or without 25 mM IsoLiPro and then subjected to immunoblotting for total tau, USP11, and GAPDH.

## LC-MS/MS detection of diGly ubiquitin signatures

Briefly, SDS was added to the sample to a final concentration of 5%, after which proteins were reduced with dithiothreitol (DTT), alkylated with iodoacetamide (IAM). Proteins were loaded onto s-traps, washed, and digested with Trypsin overnight at 37 °C. Peptides were eluted and dried with a vacuum concentrator, and then re-suspended in 10 μL of 0.1% formic acid before LC-MS/MS analysis. Peptides were separated using a 150 μm × 15 cm C18 reversed-phase-HPLC column (Thermo Fisher Scientific) on Easy-nLC 1200 (Thermo Fisher Scientific, USA) with a 66-min gradient (4–95% ACN with 0.1% formic acid) and analyzed on a hybrid quadrupole-Orbitrap instrument (Q Exactive Plus, Thermo Fisher Scientific). Full MS survey scans were acquired at 70,000 resolutions. The raw MS files were analyzed and searched against the target protein database based on the species of the samples using Byonic. Search parameters included constant modification of cysteine by carbamidomethylation and the variable modifications, methionine oxidation, protein N-term acetylation, and the addition of a GlyGly residue on Lysine. While trypsin is unable to cleave lysine residues modified by ubiquitin, trypsin cleaves the ubiquitin moiety at its C-terminus (sequence KESTLHLVLRLRGG), yielding a characteristic diGly residue that results in the addition of +114.1 Da, enabling detection by mass spectrometry.

## GST pull-down assay

Protein CHIP-231AA-GST and protein Tau-his were subjected to GST pull-down assay to verify whether protein CHIP-231AA-GST and protein Tau-his interacted. GST protein was used as a negative control. Purified CHIP-231AA-GST and Tau-his were added to individual columns with GST Magarose Beads and incubated for 4 h at 4 °C. Finally, washing buffer (1×PBS buffer with 350 mM NaCl) was used to wash the beads by running through the column. Then Western blotting was performed to determine the interaction between the protein CHIP-231AA-GST and Tau-his.

## In vitro tau ubiquitination assay

Ubiquitination assay was performed by adding Tau-his, CHIP-231AA-E3, ubiquitin-activating enzyme E1, and ubiquitin-binding enzyme E2 in the buffer (50 mM Tris-HCl, pH 7.4, 2 mM ATP,

5 mM $MgCl_2$, and 2 mM DTT). The reaction mix (30 μL) was incubated for 2 h at 37 °C. The reaction products were separated on 15% SDS-PAGE gel and subjected to immunoblot using anti-Ub antibody.

## Golgi staining and spine analyses

Golgi staining was performed by using a Rapid GolgiStain Kit (FD neurotechnologies, PK401) according to the manufacturer's instructions. Briefly, the animals were sacrificed and perfused for 5 min with PBS. All procedures were performed under dark conditions. Brains were dissected out and immersed in impregnation solution (equal volumes of Solutions A and B, containing mercuric chloride, potassium dichromate, and potassium chromate, mixed for 24 h in advance), and stored at room temperature. The impregnation solution was replaced after 24 h. After 4 weeks, brains were transferred to Solution C and stored at 4 °C for 72 h, with the solution replaced after 24 h. The brain was sectioned coronally (100 μm) using a vibrating microtome (VT 1000 s, Leica, Nussloch, Germany), and sections were mounted on gelatin-coated microscope slides with Solution C. Slides were rinsed twice in distilled water (2 min each), and then placed in a mixture of Solution D:E:distilled water (1:1:2, vol/vol/vol) for 10 min. After rinsing with distilled water, sections were dehydrated in 50%, 75, 95, and 100% ethanol four times (4 min each). Sections were cleared in xylene three times (4 min each) and coverslipped with Permount solution. The spine morphology was analyzed by Nikon microscope. The spine numbers per 10 μm of dendrite per neuron were calculated by using Image-Pro Plus 6.0, and 5 to 7 neurons per mouse (three mice per group) were used for statistical analyses.

## Aβ$_{42/40}$ ELISA

The levels of Aβ$_{42}$ and Aβ$_{40}$ were measured using the human Aβ$_{42}$ ELISA Kit (Thermo Fisher Scientific, KHB3441) and the human Aβ$_{40}$ ELISA Kit (Thermo Fisher Scientific, KHB3481), respectively, according to the manufacturer's protocol.

## Behavioral test

Animal's cognitive performance was assessed by the Y-maze test and Morris water maze (MWM) test. In the Y-maze test, mice were placed at the end of a randomly chosen arm and allowed to explore the maze for 5 min with one arm closed (designated as a novel arm). After a 2-h inter-trial interval, mice were re-introduced to the maze (in a different arm than the acquisition trial, but not the novel arm) with all arms open and allowed to explore again for 5 min. Time spent in each arm was recorded, and the percentage of time spent in the novel arm was calculated (time in the novel arm/total time in all three arms). As for the MWM test, behavioral performance was recorded in a circular tank filled with opaque water and with four reference-cued shapes affixed to the walls surrounding the tank. In the hidden platform training phase, mice were placed into the maze at 1 of 4 random points and allowed to search for the hidden platform for 60 s. If a mouse failed to find the platform within 60 s, it was guided to the platform and allowed to rest on it for 10 s. The time taken to find the platform was recorded by Smart Video Tracking Software 3.0 (Panlab, Harvard Apparatus). In the platform test phase, the hidden platform was removed.

The time spent in each quadrant and the number of crossings over the platform were scored.

## Hematoxylin and eosin (HE) staining

After embedding the major organs of the mice in paraffin, the tissues were sliced into 4-μm consecutive sections and subjected to the following staining steps according to the kit instructions (Solarbio, G1120): deparaffinization in xylene, dehydration in graded ethanol (100, 95, 80%), hematoxylin staining (5–20 min), differentiation(30 s), eosin staining (2 min), dehydration in graded ethanol, permeabilization in xylene, and mounting with neutral resin. Finally, the pathological changes in the tissues were observed under a light microscope.

## Cell viability assay

The viabilities of SH-SY5Y or HEK293 cells in each group were assessed using the Cell Counting Kit-8 (CCK-8) assay (Beyotime, C0038). A 10 μL aliquot of CCK-8 solution was added to each well and incubated for 1 h. The absorbance at 450 nm in each well was then measured using a microplate reader, and the survival rate was calculated accordingly.

## Statistical analysis

All data were collected and analyzed in a blinded manner. Data were analyzed using GraphPad Prism software (La Jolla, CA, USA). Statistical analyses were performed using a two-tailed $t$-test for two groups, or one-way ANOVA followed by Tukey's HSD test for multiple comparisons among more than two groups. All bar plots were presented as mean ± SEM. Significance was set at $P < 0.05$%.

# Results

## Synthesis and characterization of IsoLiPro

IsoLiPro was synthesized using single solvent crystallization. Equal molar amounts of lithium isobutyrate (LiIB) and L-proline (Pro) were refluxed in $n$-butanol, stirred for about 3 h, then cooled naturally to crystallize (Fig. EV1A). The yielded crystal (IsoLiPro) was collected and characterized using various analysis techniques.

Powder X-ray diffraction analysis (Fig. EV1B) revealed that the solid precipitate under n-butanol as the sole solvent showed distinct 2θ peak values: 7.942, 8.322, 11.828, 16.651, 16.884, 18.840, 20.511, 23.782, 25.086, 30.364, 31.574, 33.663, and 35.951, signifying the unique crystal structure of IsoLiPro. This pattern markedly differed from that of LiIB, Pro, or their mixture, highlighting a distinct crystalline identity of IsoLiPro.

The 1H-NMR (400 MHz, MeOD) spectrum of IsoLiPro (Fig. EV1C) displayed distinct chemical shifts, with values including δ = 4.08-4.02, 3.40-3.31, 3.30-3.21, among others, reflecting the unique hydrogen environments in the molecule. These shifts differ from those observed in Pro, lithium proline (ProLi), isobutyric acid (IB), and LiIB, indicating the lithium atom affects the electron cloud distribution of hydrogen atoms in Pro and IB in the solution.

Fourier transform infrared analysis (Fig. EV1D) revealed distinct absorption peaks, indicative of various chemical bonds in

the sample. Peaks at 2953.4 and 2780.5 cm$^{-1}$ represented N-H and -CH3 groups, respectively. The C=O group was marked by a medium-strong peak at 1598.5 cm$^{-1}$. Ester bonds showed strong absorption at 1169.6 and 1294.9 cm$^{-1}$. In addition, the peak at 548.3 cm$^{-1}$ indicated Li-O bonds. The absence of significant peak splitting suggests consistent Li-O bond lengths or energies, reflecting stable interactions between lithium and Pro or IB.

Differential scanning calorimetry analysis (Fig. EV1E) showed that IsoLIPro had a significant endothermic peak at 233.4 °C, with a sharp and single peak, indicating that IsoLiPro was stable and pure. The magnetic resonance mass spectrometry analysis (Fig. EV1F; Appendix Fig. S1A) revealed a significant absorption peak at a mass-to-charge ratio of 208.11643, indicating a relative molecular mass of 209.

The single crystal x-ray structural analysis (Appendix Table S1) of IsoLiPro highlighted a unique composition within its unit cell: four lithium cations, four IB anions, and four Pro molecules, accompanied by one crystal water molecule. The close match of the C-O bond lengths in Pro suggested an inner salt formation, rather than a co-crystal structure. Lithium ions form a tetrahedral configuration with Pro and IB, with bond lengths typical for coordination bonds. This structure confirms IsoLiPro as a lithium isobutyrate-proline coordinating salt, characterized by distinct molecular arrangement. This aligns with the chemical formula $C_9H_{15}LiNO_4$. Crystallographic data suggested monoclinic crystals with specific dimensions and angles ($\alpha = 90°$, $\beta = 91.363°$, $\gamma = 90°$; crystal column: a = 10.3529 (7), b = 10.4917 (6), c = 10.4024 (8), since a ≠ b ≠ c $\alpha = \gamma = 90°$). Although the hydrogen spectrum analysis of IsoLiPro mirrors that of a mechanical mixture of its components, further x-ray Diffraction and infrared spectroscopy analyses confirmed its unique identity, differentiating it from LiIB and Pro.

Worth noting, the two-dimensional contour map and linear standard curve revealed a distinct fluorescence absorption property of IsoLiPro in aqueous solution (Appendix Fig. S1B), suggesting the existence of non-ionized IsoLiPro. Consistently, the Tandem quadrupole mass spectrometry (QTRAP® 5500 system) (Fig. EV1G,H) and metabolomics multiple reaction monitoring-mass spectrometry (MRM-MS) (Appendix Fig. S1C) analyses of brain or serum samples from rats treated with IsoLiPro also identified the non-ionized whole molecule of IsoLiPro.

## IsoLiPro markedly reduces levels of total and phosphorylated tau in cultured cells overexpressing tau441

Considering the previously reported close interplay between USP11 and AD-related tau pathology, in our present study, we further evaluated the impacts of IsoLiPro on the levels of total tau and phosphorylated tau (p-tau). Using cell-based experiments, IsoLiPro's ability to promote tau protein degradation by inhibiting USP11 protein levels was first demonstrated. As shown in Fig. 1A,B, the levels of both USP11 and total tau proteins were significantly downregulated after IsoLiPro treatment in HEK293-hTau cells overexpressing tau441.

To further confirm the interactions between USP11 and tau and the involvement of USP11 in IsoLiPro-induced tau reduction, HEK293-hTau cells overexpressing USP11 (HEK293-hTau/USP11) was conducted. As shown in Fig. 1C,D, WB data indicated that overexpression of USP11 (USP+, HEK293-hTau/USP11) significantly upregulated the protein level of total-tau, compared to cells

without overexpression of USP11 (USP-, HEK293-hTau). Intriguingly, following 24 h treatment with IsoLiPro, a significant decrease in total-tau was observed in both HEK293-hTau and HEK293-hTau/USP11 cells, with HEK293-hTau/USP11 cells much stronger. In addition, consistent with Western blotting, IF staining suggested that, after IsoLiPro treatment, notable decreases in the fluorescence intensity of both USP11 and tau proteins were observed in SH-SY5Y cells (Fig. EV2A). Additionally, a significant reduction in their co-localizations was also observed, reinforcing the impact of IsoLiPro exposure on the interaction between these two proteins (Fig. EV2A). Therefore, IsoLiPro may promote tau degradation by reducing the USP11 level.

Subsequently, to investigate if IsoLiPro enhances tau clearance by disrupting USP11-tau stability, we conducted an immunoprecipitation (IP) analysis (Fig. 1E). Cell lysates prepared from HKE293-hTau/USP11 were immunoprecipitated using anti-tau antibody, and then total-tau and USP11 proteins were determined using Western blotting. As expected, significantly reduced interactions between cellular tau and USP11 can be observed after IsoLiPro treatment (Fig. 1E). In addition, the impaired interaction between USP11 and tau was further confirmed in vitro by IF staining, as evidenced by the weakened co-localizations of these two proteins (Fig. EV2A). To ascertain if IsoLiPro mediates tau degradation via the proteasome pathway, we introduced a proteasome inhibitor, MG132, into the study. The results indicated that IsoLiPro significantly accelerated tau degradation, but this effect was hindered upon MG132 (Figs. 1F,G and EV2B,C). This suggests that IsoLiPro likely promotes tau protein breakdown through the proteasome degradation pathway. The CHX-chase experiment also showed that IsoLiPro could accelerate the degradation of tau protein by promoting its turnover rate (Figs. 1H, I and EV2D,E).

To evaluate the time- and dose-dependence of the IsoLiPro-induced tau reduction, the total tau and p-tau levels in HEK293-hTau cells, or SH-SY5Y cells as well as iPSCs stably transfected with tau441 (iPSCs-hTau) were assessed by Western blotting. We observed that IsoLiPro treatment induced a time- and concentration-dependent decrease in both total tau and p-tau levels at multiple phosphorylation sites, consistent with the downregulation of USP11 (Figs. 1J–M and EV2F–K). Interestingly, tau mRNA level in SH-SY5Y and HEK293-hTau cells treated with IsoLiPro remained stable, also implying the possible involvement of post-translational modification-related mechanisms in IsoLiPro-induced tau reduction (Fig. EV2L).

## IsoLiPro markedly decreases USP11 protein levels both in vitro and in vivo

To further confirm the inhibitory effect of IsoLiPro on USP11, the inhibiting effect of various small molecules on USP11 protein levels were first assessed in vitro in SH-SY5Y or HEK293 cells using Western blotting, with USP11 inhibitor Mito as positive control. As shown in Figs. 2A and EV3A–D, while Mito significantly reduced USP11 protein levels after 24 h exposure, four small-molecule lithium compounds, including IsoLiPro, ProLi, LiIB, and Sal-Li-Pro, also dramatically inhibited USP11 protein levels, with IsoLiPro as the strongest. In contrast, LiCl, $Li_2CO_3$, Hom-Li-Pro, and homotuarine showed no impact on USP11 protein levels, indicating a lithium-independent mechanism. Our data also indicated that IsoLiPro reduced the USP11 protein levels in a time- and

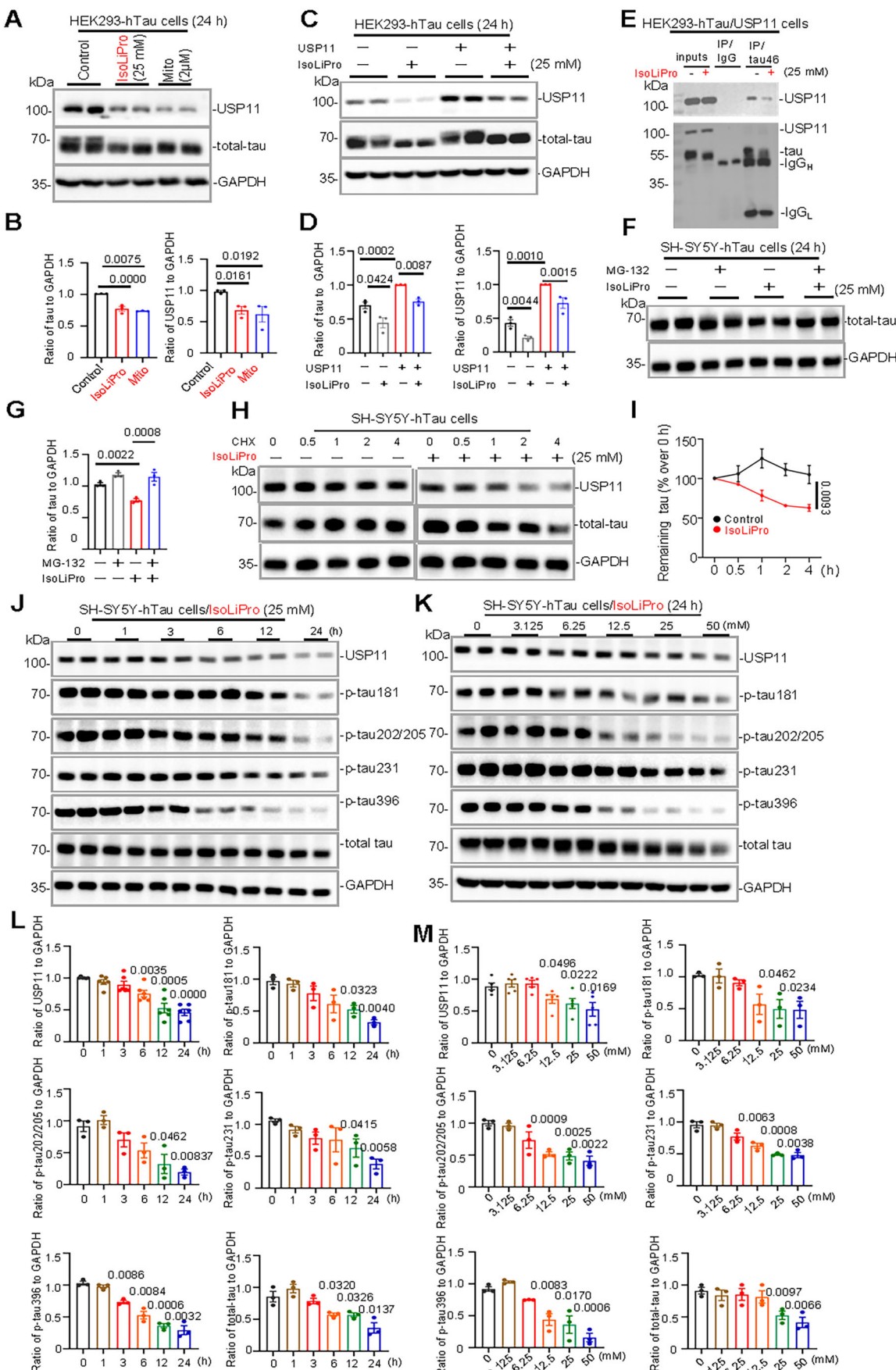

◄ **Figure 1. IsoLiPro markedly reduces levels of total and phosphorylated tau in cultured cells overexpressing wild-type full-length human tau.**

(A, B) Treatment with IsoLiPro (25 mM) and Mitoxantrone (Mito, 2 μM) significantly reduces the protein level of USP11 and total-tau protein in HEK293-hTau cells over 24 h, as measured by Western blotting. Data were represented as means ± SEM ($n = 3$ samples per group). $P$ values were calculated using multiple $t$-tests, with comparisons made against the Control group. (C, D) In HEK293-hTau cells, both transfected and untransfected with USP11, IsoLiPro (25 mM for 24 h) accelerates tau degradation, the degree of this degradation correlates with the extent of decreased USP11 protein levels. Data are represented as means ± SEM ($n = 3$ samples per group). $P$ values were calculated using one-way ANOVA followed by Tukey's HSD test. (E) Immunoprecipitation analysis reveal that IsoLiPro (25 mM for 24 h) disrupts significantly the interaction between USP11 and tau in HEK293-hTau/USP11 cells. (F, G) IsoLiPro (25 mM for 24 h) markedly accelerates tau clearance, altered by inhibition of the proteasomal (MG132, 10 μM) degradation signaling pathways in SH-SY5Y-hTau cells. Data were represented as means ± SEM ($n = 3$ samples per group). $P$ values were calculated using one-way ANOVA followed by Tukey's HSD test. (H, I) IsoLiPro markedly enhances tau turnover in SH-SY5Y-hTau cells (CHX, 100 μg/mL), as quantified in the accompanying graph. Data were represented as means ± SEM ($n = 3$ samples per group). $P$ values were calculated using two-way ANOVA, with comparisons made against the Control group. (J–M) IsoLiPro markedly reduces the levels of total and phosphorylated tau in a time- and concatenation-dependent manner in SH-SY5Y cells stably expressing wild type full-length human tau. Data were represented as means ± SEM ($n = 3$ to 5 samples per group). $P$ values were calculated using multiple $t$-tests, with comparisons made against the 0 h or 0 mM group. Source data are available online for this figure.

concentration-dependent manner (Figs. 2B,C and EV3E,F). To ascertain whether the reduction of USP11 protein levels is related to transcriptional level regulation, USP11 mRNA levels were further detected after IsoLiPro exposure. Our data showed that IsoLiPro does not significantly suppress USP11 mRNA expression (Fig. EV3G). Additionally, enzyme activity assay results indicated that IsoLiPro also failed to affect USP11 enzyme activity (Fig. EV3H).

We further confirmed the impacts of IsoLiPro on USP11 protein levels in vivo. IHC staining and Western blotting on brain tissues of transgenic AD mice (both 3xTg and 5xFAD) showed that, after treatment with IsoLiPro, the protein levels of USP11 were significantly reduced in the cortex and hippocampus, compared with double-distilled water-treated (ddH$_2$O) control mice (Figs. 2F–H and EV3I,J). Consistent with in vitro data, chronic Li$_2$CO$_3$ treatment in 5xFAD mouse failed to induce a significant reduction of USP11 protein level (Fig. EV3I,J).

Additionally, the specificity of USP11 protein level inhibition by IsoLiPro was determined by examining the protein level of USP25, another AD-related member of the USPs family. As expected, while Mito non-selectively and significantly reduced USP25 protein levels after 24 h exposure, IsoLiPro at 12.5 mM showed no significant impact on USP25 protein level (Fig. 2D,E), suggesting IsoLiPro may have preferential inhibition, particularly towards USP11.

## IsoLiPro shows a strong binding affinity to USP11

To further investigate the mechanism by which IsoLiPro inhibits USP11 protein, molecular docking of USP11 (PDB ID: 4MEL) with IsoLiPro was conducted using Autodock Vina and visualized via PyMOL software. As shown in Fig. EV4A,B a significant binding site of USP11 for IsoLiPro was identified at conserved 149-152 (VEVY) motif, a notable characteristic of the N-terminal "domain present in USPs" and "ubiquitin-like" (UBL) domain (DU) interface of USP11. This motif plays a critical role in facilitating protein–protein interactions. In contrast to IsoLiPro, our molecular docking further indicated that lithium carbonate (Li$_2$CO$_3$), LiIB, Pro, and mitoxantrone (Mito) shows distinct binding residues on USP11 (out of VEVY motif) (Appendix Fig. S2A–E). Additionally, IsoLiPro also demonstrated some extent of binding to other USPs, but the binding affinities are lower than USP11 (Appendix Fig. S2F,G).

To further explore the binding affinity of IsoLiPro with USP11, molecular dynamics (MD) simulations and binding free energy analysis (MM-PBSA) were conducted. Our results (Fig. EV4C;

Appendix Table S2) indicated that electrostatic energy (ΔEelec: −104.03 kcal/mol) and van der Waals (ΔEvdw: −13.36 kcal/mol) primarily drive the binding of IsoLiPro to USP11, which were much stronger than other two lithium compounds, Li$_2$CO$_3$ and LiIB. However, this binding energy between IsoLIPro and USP11 was offset by the solvation-free energy (PB-SOL: 96.85 kcal/mol), resulting in an overall lower total interaction affinity (ΔG(Total) = −20.54) than Li$_2$CO$_3$ and LiIB (ΔG (Total) = −26.56 and −30.49, respectively). Pro showed the weakest binding affinity among the tested compounds.

The intermolecular interaction between IsoLiPro and recombinant USP11 was further evaluated in vitro by microscale thermophoresis (MST) assay. Our data revealed that IsoLiPro can strongly bind to USP11 with a dissociation constant (Kd) of 4.3394E-05 M (Fig. EV4D). This low Kd value signifies a strong affinity between IsoLiPro and USP11, underscoring its potential in therapeutic applications.

The differences in binding to USP11 among these molecules were further elucidated through analysis of MD parameters, including RMSD, RMSF, Rg, and SASA. RMSD calculations over 100 ns of MD simulation revealed varied stability among the systems. As shown in Fig. EV4E, while the RMSD value of IsoLiPro rapidly increased to a stable level around 1.1 Å after an initial spike at 5 ns, the RMSD of Li$_2$CO$_3$ system showed a significant increase to 1.6 Å at 19 ns, stabilizing near 1.5 Å after some fluctuations. RMSD of LiIB stabilized around 1.4 Å by 25 ns, whereas Pro's RMSD shot up to ~1.7 Å at 7 ns, suggesting less structural stability. These results highlight the superior structural stability of IsoLiPro compared to other tested compounds. RMSF analysis revealed that IsoLiPro uniquely maintained low fluctuation throughout the simulation, contrasting with other systems, especially the USP11/Li$_2$CO$_3$ complex (Fig. EV4F). This trend suggests IsoLiPro effectively constrains protein fluctuations, indicating a more restricted and stable protein structure compared to others. Rg values offer insights into protein folding dynamics, with higher Rg indicating a less compact protein–ligand complex. Our data indicated that the complex of IsoLiPro and USP11 exhibited the lowest Rg, suggesting the tightest binding affinity (Fig. EV4G). SASA is instrumental in predicting conformational changes, analyzing interactions between complexes and solvents, and estimating the number of surface and hydrophobic core residues in proteins. Our data indicated that, during a 100 ns simulation, the SASA value of IsoLiPro was decreased significantly at 20 ns and maintained a relatively lower level compared to that of other

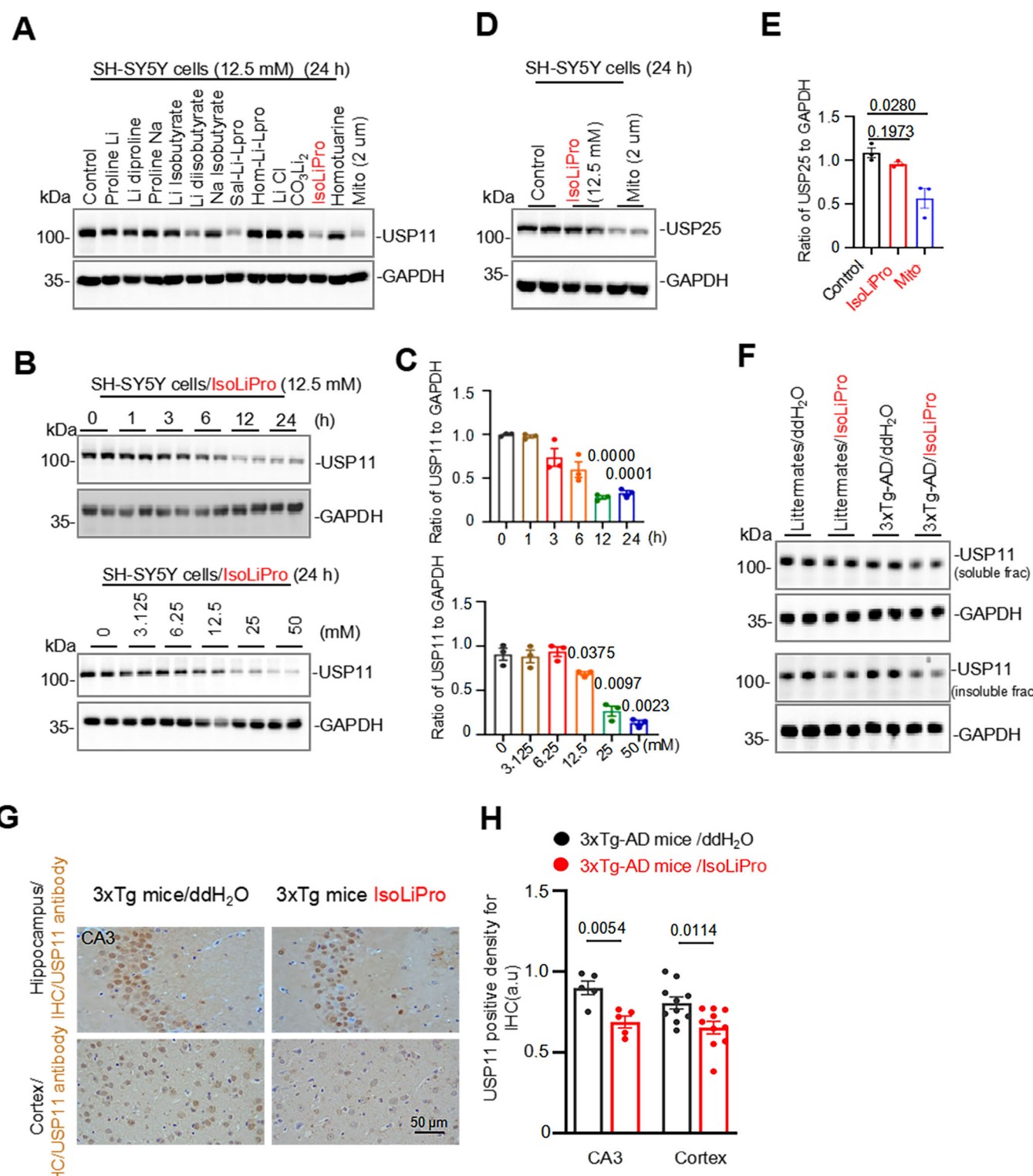

complexes, suggesting the formation of a more compact and stable complex (Fig. EV4H). Taken together, all these results indicate that while Li$_2$CO$_3$ and LiIB have higher binding energy than IsoLiPro. However, the binding sites of IsoLiPro are very different from other compounds. IsoLiPro can directly bind to the VEVY motif of USP11. In addition, during the binding process, IsoLiPro exhibits less variation in molecular structure, leading to a tighter and more effective binding. This highlights IsoLiPro's distinctiveness in terms of binding efficiency and stability compared to other lithium-containing compounds.

**Figure 2.  IsoLiPro markedly decreases USP11 protein level in SH-SY5Y cells and the mice brain.**

(A) Representative blots showing the impact of various small molecule drugs on USP11 protein levels in SH-SY5Y cells. (B, C) IsoLiPro induces a time- and concentration-dependent inhibition of USP11 in SH-SY5Y cells measured by Western blotting. Data were represented as means ± SEM ($n = 3$ samples per group). $P$ values were calculated using multiple $t$-tests, with comparisons made against the 0 h or 0 mM group. (D, E) Comparative analysis reveals IsoLiPro's selective inhibition of USP11 expression over USP25. Data were represented as means ± SEM ($n = 3$ samples per group). $P$ values were calculated using multiple $t$-tests, with comparisons made against the Control group. (F) Oral administration of IsoLiPro in 9.5-month-old 3xTg-AD mice for 16 weeks substantially reduced USP11 protein level in both soluble and insoluble fractions, as evidenced by Western blotting. (G, H) IsoLiPro markedly decreased USP11 levels in the hippocampus and cortex of 3xTg-AD mice, confirmed by IHC with a USP11-specific antibody. Data are represented as means ± SEM ($n = 3$ to 5 mice per group, with two cortical sections analyzed from each mouse). $P$ values were calculated using a two-tailed $t$-test, with comparisons made against the 3xTg-AD mice/ddH$_2$O. Source data are available online for this figure.

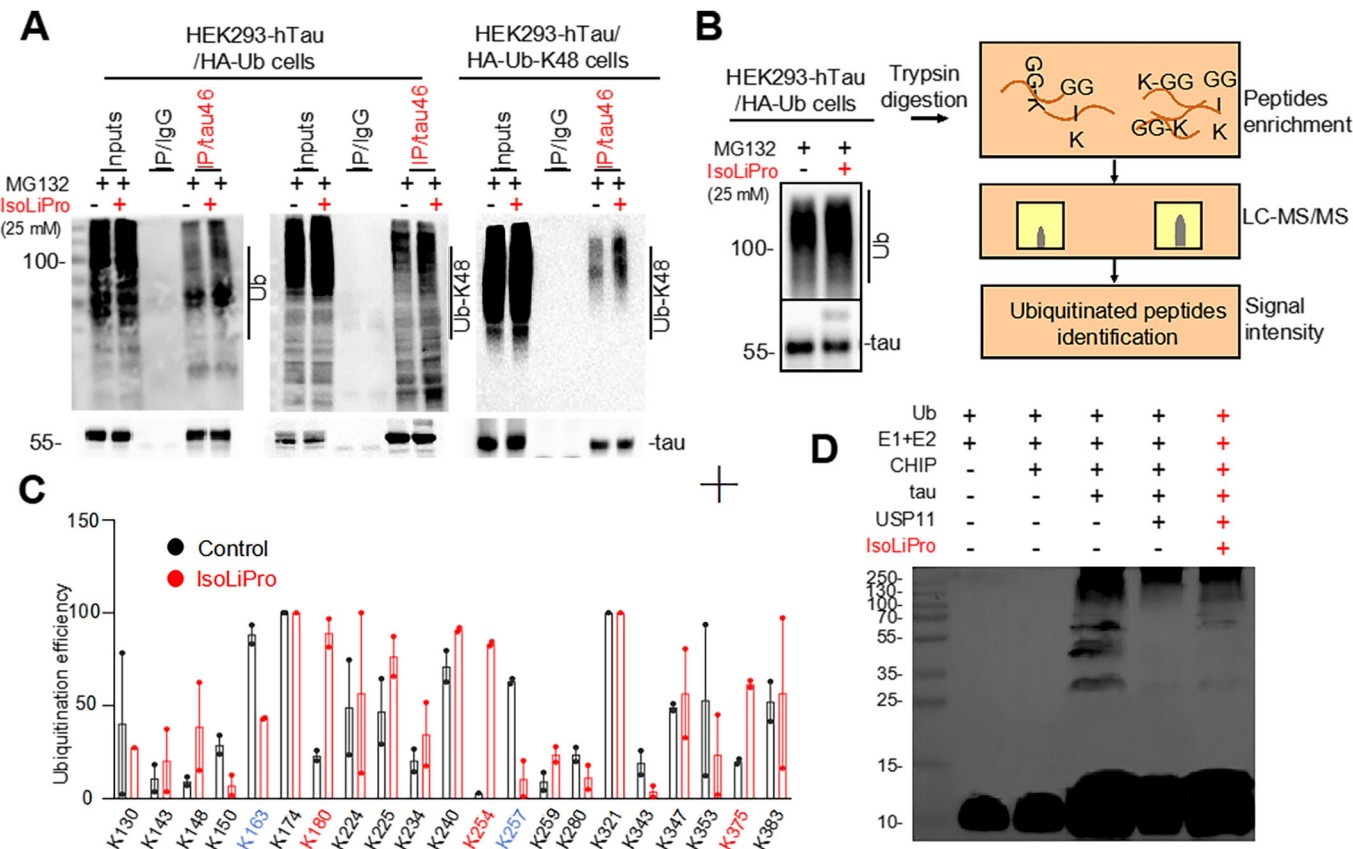

**Figure 3.  IsoLiPro alters tau ubiquitination levels in HEK293-hTau cells.**

(A) IP analysis show the increased poly-ubiquitination (panel left), Ubiquitin-K48 (panel middle and right) conjugates and tau in HEK293-hTau cells treated with 25 mM IsoLiPro and 10 µM MG132. (B) Schematic of representation of the detection of diGly-ub peptide signatures via LC-MS/MS. (C) Quantification analysis of diGly-ub signatures remaining on tau lysine in tau-pulldown samples (IP: tau) from HEK293-hTau cells transfected with HA-ubiquitin, identified via LC-MS/MS. Data were represented as means ± SEM ($n = 2$ samples per group). (D) Representative blots showing IsoLiPro inhibits the deubiquitination of CHIP- ubiquitinated recombinant tau by recombinant USP11 protein. Source data are available online for this figure.

## IsoLiPro markedly increases tau ubiquitination levels in HEK293-hTau cells

USP11 regulates the deubiquitination of tau protein, leading to its accumulation and consequently exacerbating the progression of tau pathology. Inhibiting USP11 can enhance the ubiquitination level of tau, therefore promoting its degradation by the proteasomal or lysosome-autophagy system. This approach could alleviate cognitive impairments in dementia mouse models, suggesting that effective suppression of USP11 function by inhibitors could provide a potential therapeutic strategy for AD.

To ascertain whether IsoLiPro promotes tau degradation by enhancing its ubiquitination, IP analysis was used to measure the ubiquitination level of tau in HEK293-hTau cells transfected with HA-Ub (HEK293-hTau/HA-Ub) or HA-Ub-K48 (HEK293-hTau/HA-Ub-K48), after 25 mM IsoLiPro treatment. Our results indicated a significant increase in the overall ubiquitination level of tau, primarily through Ub-K48 linkage (Fig. 3A), while Ub-K63 linkage levels remained largely unchanged (Appendix Fig. S3A). Ub-K48, primarily linked to proteasomal degradation, contrasts with Ub-K63, which is linked to the slower clearance of misfolded or insoluble proteins via the autophagy-lysosome pathway. Thus,

IsoLiPro predominantly engages in the ubiquitin-proteasome degradation pathway for tau protein.

Following IP and SDS electrophoresis, the samples were trypsinized, enriching K-ε-GG peptides for LC-MS/MS analysis to assess the impacts of IsoLiPro on the overall ubiquitination level and amino acid ubiquitination efficiency of tau. The results showed that IsoLiPro treatment not only promote the ubiquitination but also alters the ubiquitination sites on lysine residues of tau (Fig. 3B,C; Appendix Fig. S3B,C). Through analyzing ubiquitinated sites on tau protein, changes in ubiquitination efficiency at specific amino acid residues were noted. For instance, ubiquitination at K180, K254, and K375 increased significantly, while it decreased at K163 and K257. Despite these site-specific changes, the overall total ubiquitination level was increased. This suggests that the ubiquitination degradation mechanism of tau involves not just increased ubiquitination levels, but also a shift in ubiquitination sites. The functional implications of these site-specific ubiquitination changes warrant further investigation.

To further understand IsoLiPro's role in tau ubiquitination, we simulated recombinant tau protein ubiquitination in vitro, finding recombinant tau protein could bind and interact with GST-tagged CHIP-231AA (Appendix Fig. S3D). Subsequent experiment involved Tau-his, protein CHIP-231AA-E3, ubiquitin, E1 and E2, confirming ubiquitnation of tau-his. Tau ubiquitination was diminished by the addition of recombinant USP11 but partially restored by IsoLiPro, suggesting IsoLiPro promotes tau ubiquitination and degradation by inhibiting USP11 (Fig. 3D).

## IsoLiPro markedly reduces total- and phospho-tau levels in 3xTg-AD and 5xFAD mice

To investigate IsoLiPro's role in promoting tau clearance in vivo, 9.5-month-old 3xTg mice were orally treated with IsoLiPro (560 mg/kg body weight for 16 weeks, gavage). This treatment significantly reduced total tau and p-tau levels in the brains, as assessed by IHC/IF staining (Fig. 4A–F). Consistently, Western blotting data also indicated that both soluble and insoluble forms of USP11, total tau, and p-tau were significantly reduced after IsoLiPro treatment (Fig. 4G,H). Moreover, similar as 3xTg-AD mice, after treatment of IsoLiPro to 10-week-old 5xFAD mice for 12 weeks, the IHC staining (Fig. EV5A–D) and Western blotting (Fig. EV5E–G) results also revealed that IsoLiPro treatment reduced USP11, total tau and p-tau levels in 5xFAD mice. However, no correlation was evident between the reduced levels and drug concentrations. There was no significant difference between low/medium concentrations, except that the inhibition effect was most pronounced at high concentrations (Fig. EV5G).

## Mitigation of Aβ pathology and associated gliosis by IsoLiPro in 5xFAD mice

AD often involves neuroinflammation, where Aβ and tau lesions exacerbate pro-inflammatory factor release and glial cells (microglia and astrocytes) proliferation. IHC and IF staining were performed to examine microgliosis and astrogliosis using the established markers Iba1 and GFAP, respectively. After chronic treatment with IsoLiPro, significant reductions in Iba1 (Fig. 5A,C) and GFAP (Fig. 5B,D) levels were observed in the hippocampus of 5xFAD mice, but not in 3xTg-AD mice (Appendix Fig. S4A–D),

suggesting the effectiveness of IsoLiPro in alleviating AD-related neuroinflammation.

To address the amyloid hypothesis, which attributes the pathogenesis of AD to toxic Aβ accumulation, we examined the effect of IsoLiPro on Aβ lesions in 5xFAD mice. IsoLiPro treatment significantly reduced the Aβ burden in the hippocampus of mice compared to that in the hippocampus of untreated 5xFAD control mice (Figs. 5E,F and EV5H,I), as evidenced by $A\beta_{17\text{-}24}$-specific 4G8 antibody IHC results. Consistently, ELISAs also indicated that IsoLiPro treatment significantly reduced the $A\beta_{42}$ and $A\beta_{42/40}$ levels in the brain lysates of 5xFAD mice (Fig. 5G,H).

## IsoLiPro ameliorates synaptic plasticity and spatial memory deficits in 3xTg-AD and 5xFAD mice

To evaluate synaptic loss, we conducted assays to measure the protein expression levels of key neuronal synaptic markers, i.e., post-synaptic density-95 (PSD-95) and synapsin 1 (SYN-1). The treatment of 3xTg-AD and 5xFAD mice with IsoLiPro significantly increased the levels of SYN-1 and PSD95, as assessed by IHC staining (Fig. 6A–D) and Western blotting (Fig. 6E,F). These results indicate an improvement in synaptic function in AD model mice following IsoLiPro treatment.

Dendrites and axons, which are crucial components of neurons, influence cognitive function, and alterations in these components exacerbate cognitive dysfunction. Golgi staining, which visualizes neuronal structure, revealed that the administration of IsoLiPro significantly increases the number and density of dendritic spines (Fig. 6G,H). This suggests that IsoLiPro effectively improves neuronal plasticity impairments commonly seen in AD models, thereby potentially enhancing cognitive function.

To assess whether IsoLiPro could improve cognitive deficits in AD mice, we utilized the MWM test to evaluate spatial cognition in 16-month-old female 3xTg-AD mice. After 4 days of training, the time taken by 3xTg-AD mice to locate the platform was measured. Post-IsoLiPro treatments, there was a noticeable improvement in the time taken by the 3xTg-AD mice to find the platform, indicating an enhancement in their spatial memory capabilities (Fig. 7A,C). The Y-maze experiment, assessing the natural exploratory behavior and spatial novelty recognition of mice, revealed notable cognition improvements in 3xTg-AD mice after chronic IsoLiPro administration. The residence time and exploration distance of IsoLiPro-treated 3xTg-AD mice in the new arm were significantly longer than that in control animals (Fig. 7B,D).

## IsoLiPro exhibits good pharmacokinetics without obvious toxicity

To evaluate tissue distribution and pharmacokinetic parameters, after a single oral administration of IsoLiPro or $Li_2CO_3$ at a dose of 9.29 mg/kg in SD rats, blood and tissue samples were collected at 1, 4, 24, and 48 h post-administration. The results showed that both IsoLiPro and $Li_2CO_3$ were detectable in most tissues and plasma (indexed by lithium), and the lithium concentration peaked at 1 h or 4 h after administration and then decreased (Appendix Fig. S5A,B). Drug concentration–time curves showed that the exposure concentration of IsoLiPro in brain tissue was significantly higher than that of $Li_2CO_3$ (Appendix Fig. S5C), and in plasma and kidney tissues, $Li_2CO_3$ quickly reached $C_{max}$, while IsoLiPro slowly

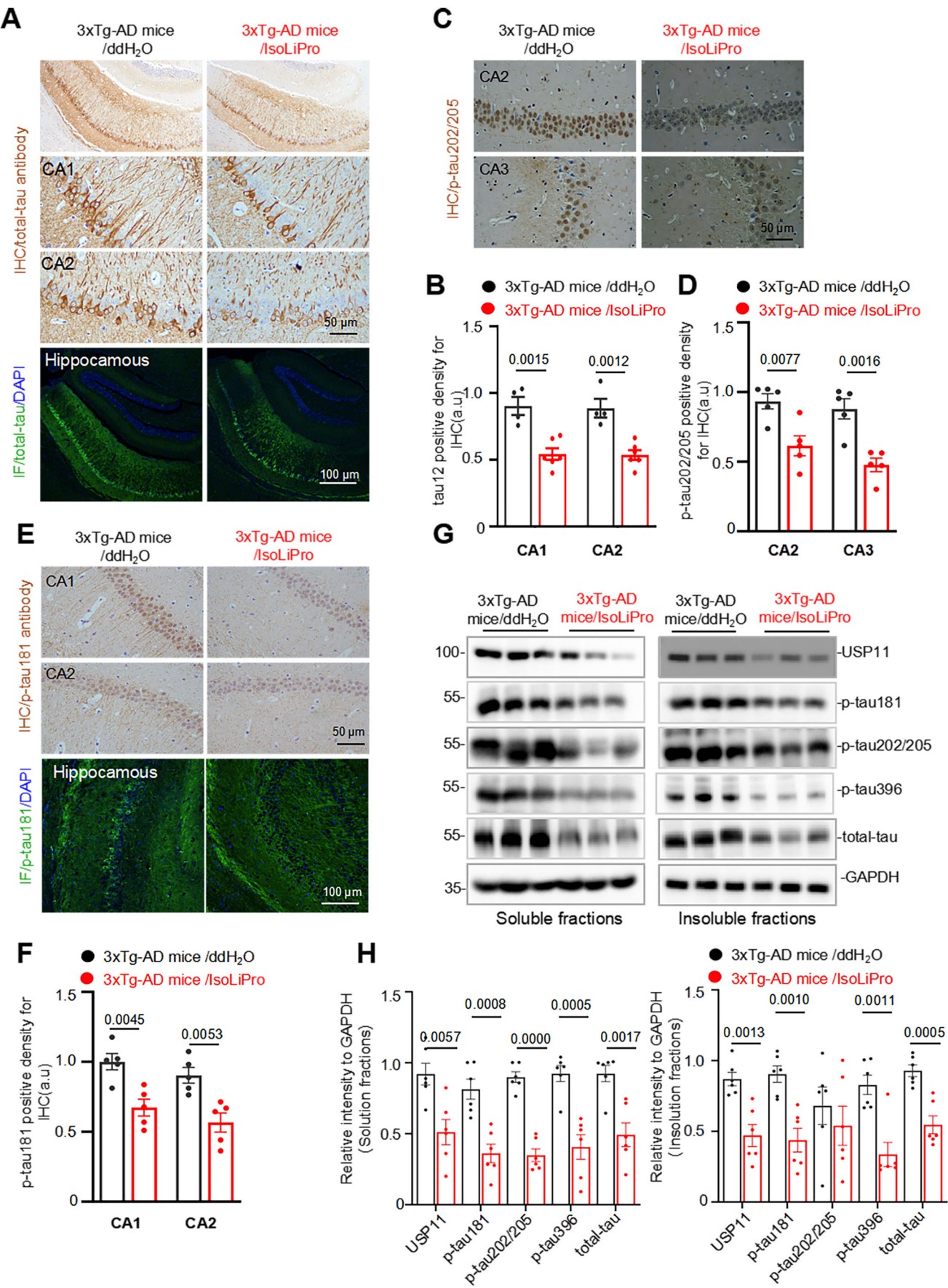

**Figure 4. IsoLiPro markedly induces tau clearance in the brain of 3xTg-AD and 5xFAD mice.**

(A–F) Administration of IsoLiPro Orally (560 mg/kg) to 9.5-month-old 3xTg-AD mice for 16 weeks significantly decreased total-tau and p-tau (p-tau181, p-tau202/205) levels in hippocampal measured by IHC/IF. Data were represented as means ± SEM ($n = 4$ to 6 mice per group). $P$ values were calculated using a two-tailed $t$-test, with comparisons made against the 3xTg-AD mice/ddH$_2$O. (G, H) IsoLiPro significantly decreased total-tau and the tau phosphorylated at multiple-doses AD-related sites (soluble and insoluble fractions) in 3xTg-AD mice measured by Western blotting. Data were represented as means ± SEM ($n = 6$ mice per group). $P$ values were calculated using a two-tailed $t$-test, with comparisons made against the 3xTg-AD mice/ddH$_2$O. Source data are available online for this figure.

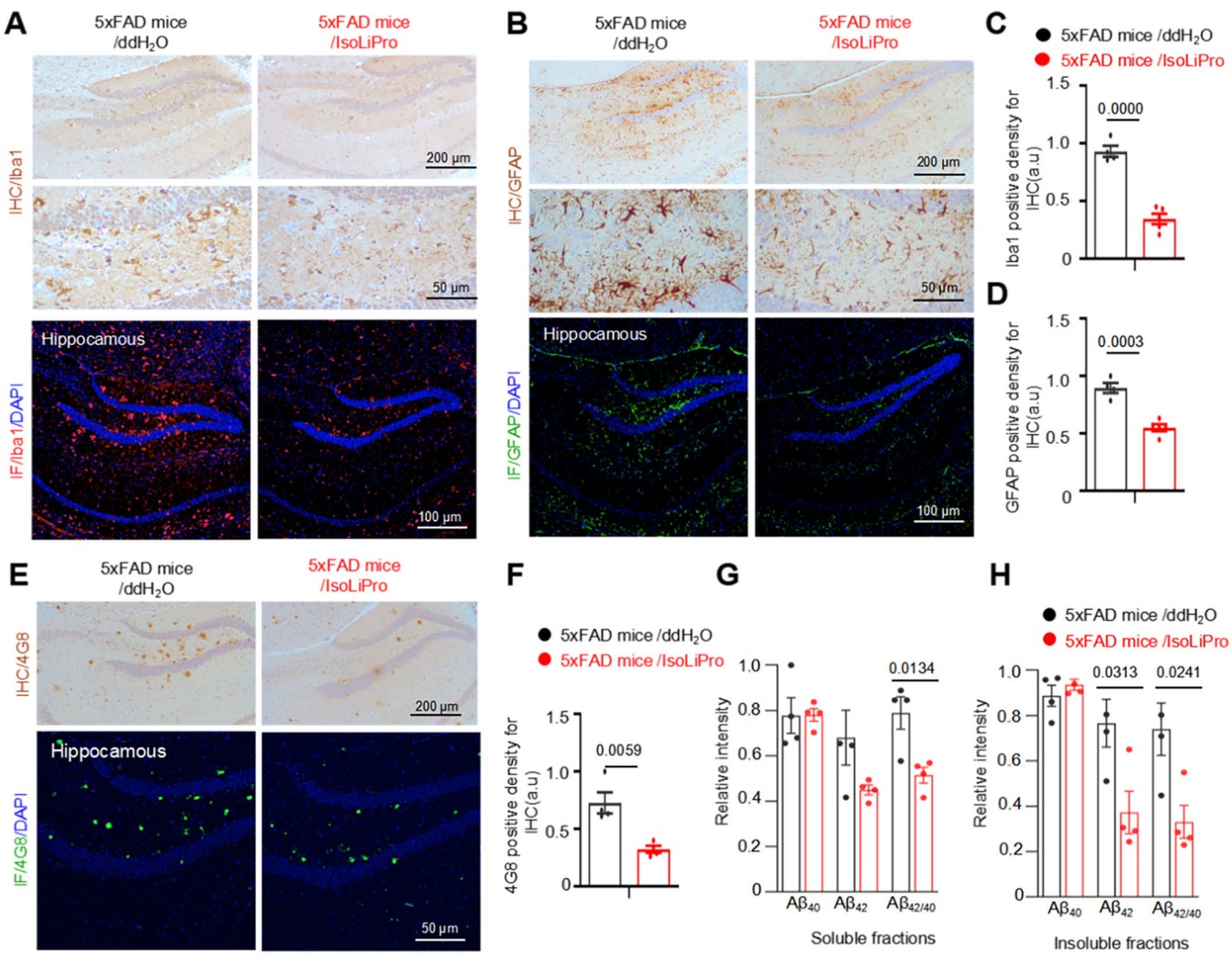

**Figure 5. IsoLiPro markedly reduces gliosis and Aβ burden in the hippocampus of 5xFAD mice.**

(A–D) Representative images of Iba1 (A, C) and GFAP (B, D) measured by IHC/IF in the hippocampus of 5xFAD mice. Data were represented as means ± SEM ($n = 4$ to 6 mice per group). $P$ values were calculated using a two-tailed $t$-test, with comparisons made against the 5xFAD mice/ddH$_2$O. (E, F) Representative IHC/IF images reveal a significant reduction in 4G8-positive amyloid plaques in the hippocampi of 5xFAD mice following IsoLiPro treatment. Data were represented as means ± SEM ($n = 4$ to 6 mice per group). $P$ values were calculated using a two-tailed $t$-test, with comparisons made against the 5xFAD mice/ddH$_2$O. (G, H) Quantification of Aβ$_{40}$, Aβ$_{42}$, and Aβ$_{42/40}$ ratio in brain homogenates of 5xFAD mice following IsoLiPro treatment. Data were represented as means ± SEM ($n = 4$ to 6 mice per group). $P$ values were calculated using a two-tailed $t$-test, with comparisons made against the 5xFAD mice/ddH$_2$O. Source data are available online for this figure.

increased and finally reached the same concentration as Li$_2$CO$_3$ (Appendix Fig. S5D,E).

To evaluate the chronic toxicity of IsoLiPro over a period of 16 weeks at high concentrations (560 mg/kg), we conducted HE staining on major organs, including the liver, kidney, heart, and lung, to detect any potential pathological damages or adverse effects. The results revealed no significant morphological changes or damage in these tissues following IsoLiPro administration (Appendix Fig. S6A). Throughout the treatment period, the monitoring of the body weight of mice showed no significant

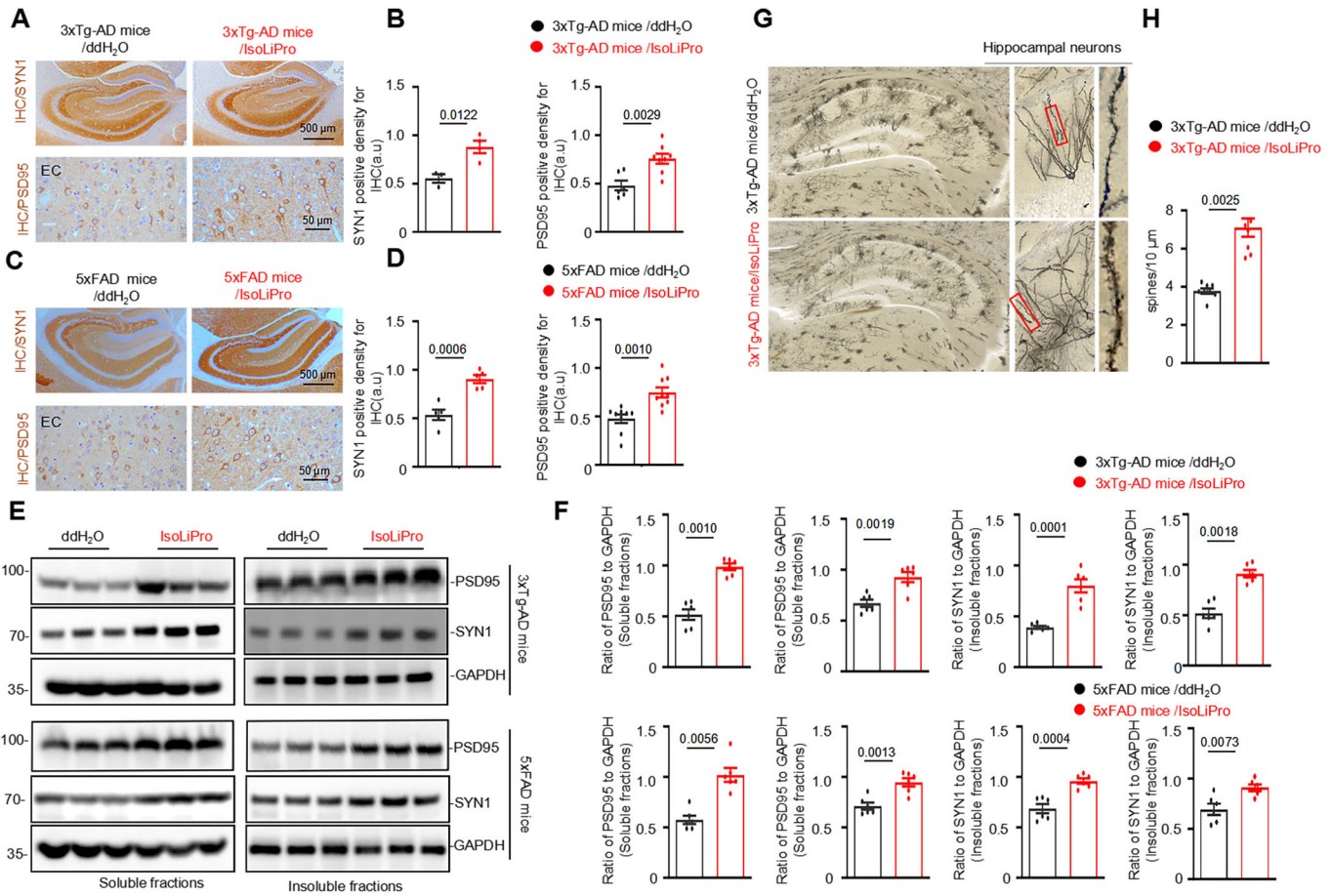

**Figure 6. IsoLiPro ameliorates synaptic plasticity deficits in the brain of 3xTg-AD and 5xFAD mice.**

(A–D) Representative images from both 3xTg-AD (A, B) and 5xFAD mice (C, D) show the enhanced expression of synaptic markers SYN-1 and PSD95, indicative of improved synaptic function. Data were represented as means ± SEM ($n = 4$ to 6 mice per group, with two cortical sections analyzed from each mouse). $P$ values were calculated using a two-tailed $t$-test, with comparisons made against the 3xTg-AD mice/ddH$_2$O or 5xFAD mice/ddH$_2$O. (E, F) Western blotting analysis confirms that IsoLiPro improves the protein levels of SYN-1 and PSD95 in both 3xTg-AD and 5xFAD mice. Data were represented as means ± SEM ($n = 4$ to 6 mice per group). $P$ values were calculated using a two-tailed $t$-test, with comparisons made against the 3xTg-AD mice/ddH$_2$O or 5xFAD mice/ddH$_2$O. (G, H) IsoLiPro treatment significantly increased spine density in the brain of 3xTg-AD mice, as measured by Golgi-cox staining. Data were represented as means ± SEM ($n = 4$ mice per group, with two sections analyzed from each mouse). $P$ values were calculated using a two-tailed $t$-test, with comparisons made against the 3xTg-AD mice/ddH$_2$O. Source data are available online for this figure.

abnormalities (Appendix Fig. S6B). These observations indicated that IsoLiPro was well-tolerated at the administered dose without significant toxicity in the evaluated organs, which were further confirmed by the data of CCK-8 assay in vitro (up to 50 mM) (Appendix Fig. S6C).

## Discussion

Increasing evidence indicates that the intracellular accumulation of tau protein plays a crucial role in neurodegeneration and memory deficits in AD and other related neurodegenerative diseases, collectively known as tauopathies (Congdon et al, 2023). Consequently, the clearance and degradation of these abnormal proteins have heightened interest in the study of tau pathologies (Goldberg, 2003). Once these proteins are ubiquitinated, they are degraded through the UPS and ALP, which is essential for maintaining

protein homeostasis (Liu et al, 2022). Ubiquitination, a dynamic and reversible PTM, is regulated by DUBs that detach ubiquitin chains, crucial for maintaining protein stability and signal transduction (Zheng et al, 2022). USP11, an important member of DUBs, has recently been identified to exacerbate tau pathology by enhancing tau protein stability and accumulation, thereby hindering its degradation and hastening AD progression (Yan et al, 2022). Moreover, a recent study has also reported that depletion of USP11 homolog H34C03.2 in Caenorhabditis elegans triggers autophagy hyperactivation and alleviates human Aβ$_{42}$ aggregation-induced paralysis (Basic et al, 2021). Therefore, the discovery of potent USP11-targeting drugs, either blocking enzyme activity or lowering protein levels, is crucial for AD treatment.

IsoLiPro was originally designed and synthesized as a novel organic lithium compound and therapeutic candidate for AD, considering the previously reported multi-targeting benefits of lithium on cognition and AD pathology, including decreased Aβ

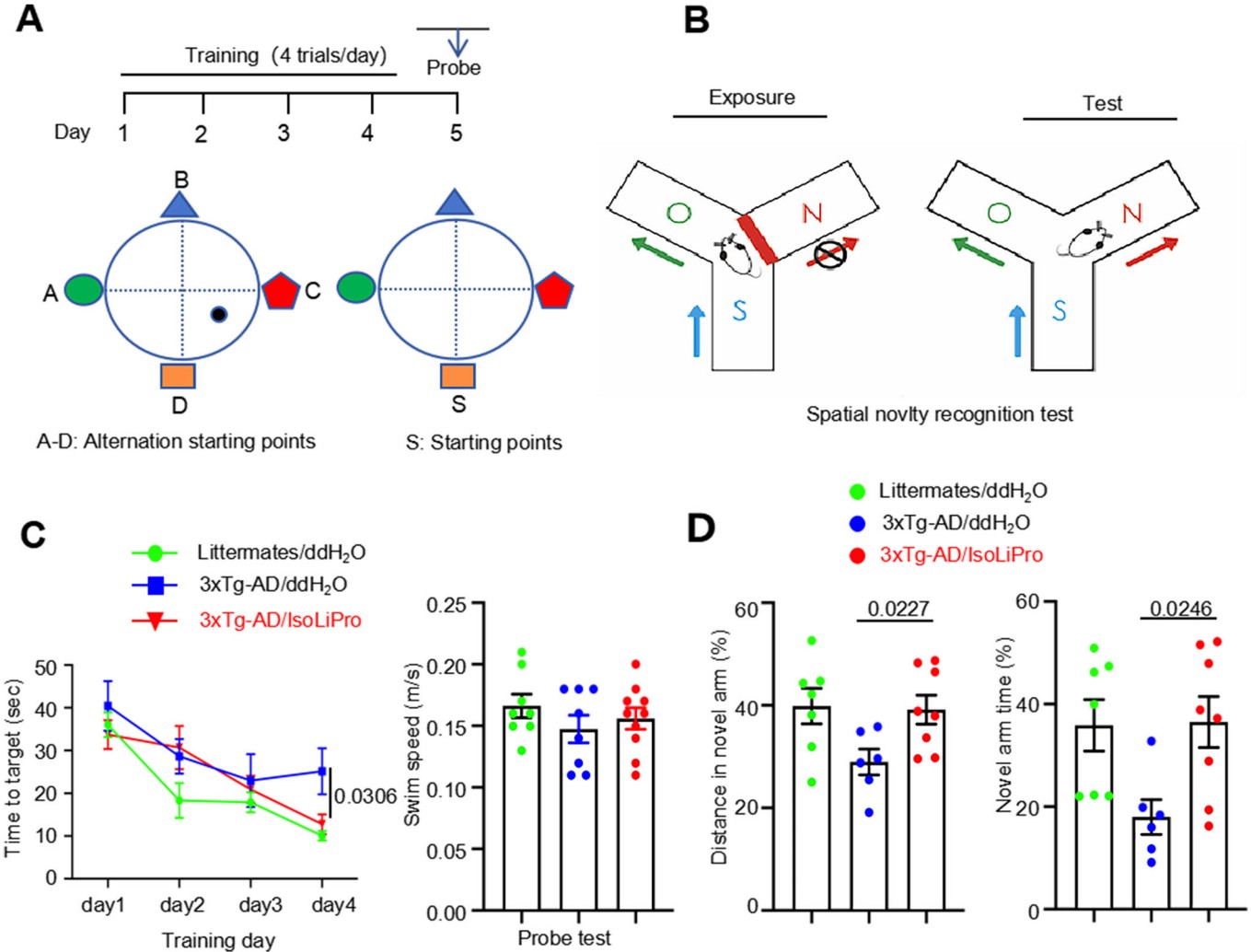

**Figure 7. IsoLiPro ameliorates spatial memory impairment in 3xTg-AD mice.**

(A, B) Schematic representations of the Morris water maze (MWM) training (A) and Y-maze blocked arm assessment (B). (C) Quantification analysis of the time taken to locate the hidden platform during the training, complemented by swim speed (m/s for 1 min) in 3xTg-AD mice. Data were represented as means ± SEM ($n = 5$ to 8 mice per group). $P$ values were calculated using a two-tailed $t$-test, with comparisons made against the 3xTg-AD mice/ddH$_2$O. (D) Statistical assessment of the time spent and distance traveled in the novel arm of the Y-maze by 3xTg-AD mice. Data were represented as means ± SEM ($n = 5$ to 8 mice per group). $P$ values were calculated using one-way ANOVA followed by Tukey's HSD test, with comparisons made against the 3xTg-AD mice/ddH$_2$O. Source data are available online for this figure.

production and tau phosphorylation (Zhang et al, 2011). The synthesis of IsoLiPro mainly aims to resolve the challenges of the clinical usage of lithium, such as the narrow lithium treatment window and high toxic side effects (Wikström et al, 2023). Isobutyric acid is selected mainly due to its widely proven application in pharmaceuticals, which can serve as an acid-base regulator, maintaining drug stability and appropriate pH value; it can also enhance drug solubility, absorption, and bioavailability, while its lower gastrointestinal irritation helps reduce common side effects such as gastrointestinal reactions. Interestingly, our preliminary data found that, despite the Aβ and p-tau ameliorations, IsoLiPro also significantly decreased total tau level.

Given the recently reported pivotal roles of USP11 in tau pathology in AD (Yan et al, 2022), we further determined the potential docking of IsoLiPro to USP11 and its inhibiting effects on USP11 activity and protein levels. Fortunately, docking assays

further revealed direct binding of IsoLiPro with USP11, and cell-based experiments indicated that IsoLiPro decreases USP11 protein level and enhances tau ubiquitination level, which might contribute to the downregulated total tau. However, our data clearly indicated that IsoLiPro decreased the protein level of USP11 without affecting its mRNA expression and catalytic activity. These findings suggest that IsoLiPro may impact USP11 function primarily through mechanisms such as regulating protein stability, metabolism, post-translational modifications, or protein–protein interactions, rather than directly inhibiting its enzymatic activity.

Molecular docking and MD simulation experiments also revealed that the binding mechanism of IsoLiPro to USP11 differs from that of Li$_2$CO$_3$, LiIB, and proline. The divergence primarily stems from the alterations IsoLiPro undergoes when it forms a coordinate salt, including modifications in its molecular structure, size, and spatial configuration, as well as shifts in its chemical

properties like polarity, charge distribution, and the equilibrium between hydrophilicity and hydrophobicity. Investigating the specific degradation mechanism of USP11 will be a focus of future research. Given that many members of the DUB family are highly conserved evolutionarily and share significant sequence similarity (Schauer et al, 2020), finding specific inhibitors presents a significant challenge. In our present study, while IsoLiPro inhibits USP11, it has little effect on USP25 protein levels, suggesting a specific inhibition of IsoLiPro on USP11.

In our present study, IsoLiPro but not lithium decreases the USP11 protein level, indicating the whole molecule IsoLiPro might be responsible for its impacts on USP11. Consistently, in its solvated state, IsoLiPro predominantly functions in its complete molecular configuration with a molecular weight of ~209. Furthermore, our findings in the mouse brain indicate that IsoLiPro is capable of efficiently penetrating the blood–brain barrier while retaining its molecular integrity.

USP11 is known to play a crucial role in regulating various diseases by stabilizing a range of substrates (Rong et al, 2022; Zhu et al, 2023). In our study, IsoLiPro, serving as an effective inhibitor of USP11, significantly enhances the ubiquitination level of tau, thereby promoting its degradation. Notably, IsoLiPro's effect was predominantly on ubiquitination involving the Ub-K48 linkage, with the Ub-K63 linkage remaining largely unaffected. The Ub-K48 modification plays a key role in the swift proteasomal clearance of soluble proteins, while the Ub-K63 modification is linked to the slower clearance of misfolded or insoluble proteins via the autophagy-lysosome pathway (Nathan et al, 2013). This suggests that the action of IsoLiPro may mainly accelerate the clearance of soluble tau monomers/oligomers rather than insoluble tau aggregates.

Further LC-MS/MS analysis revealed changes in tau ubiquitination sites after IsoLiPro treatment. The ubiquitination efficiency on specific amino acids (K180, K254, and K375) significantly increased, while it decreased at other sites (K163 and K257). These findings reveal the complexity of tau protein regulation, involving the transfer and changes in efficiency of ubiquitination sites. Further studies are still needed to understand how IsoLiPro regulates the stability of tau protein by altering specific ubiquitination sites and the impact of these changes on tau function and pathological accumulation.

Tau protein undergoes PTMs such as phosphorylation, ubiquitination, and acetylation, which alter its charge, hydrophobicity, and conformation, thereby impacting its function, protein–protein interactions, and aggregation (Ye et al, 2022). Under non-pathological conditions, tau protein phosphorylation is a prerequisite for its ubiquitination, promoting degradation in a physiological state, while leading to aggregation in pathological states. Ubiquitination by itself does not adequately generate polyubiquitinated tau species for effective proteasomal, but when combined with phosphorylation, it yields tau species that are more easily processed, aiding cells in managing stress (Kim et al, 2021). This phenomenon helps cells maintain intracellular tau concentration below a critical threshold. In our experiments, we transfected tau441 plasmids, with an expected molecular weight of approximately 46 KD (Corsi et al, 2022; Xia et al, 2021). Surprisingly, we observed multiple bands around 70 KD in HEK293/hTau when the expression level of tau is high enough, likely indicative of tau proteins altered by post-translational modifications. Our results

show that following the inhibition of USP11 by IsoLiPro, the degradation of these approximately 70 KD tau proteins was first promoted. Unlike phosphorylation, precedent acetylation did not affect the immunoblotting patterns of tau ubiquitylation (Kim et al, 2021). Hence, combining DUB inhibitors with enzymes that modulate PTMs, such as kinase or phosphatase inhibitors, may provide more effective prarmaceutical strategies for disease management. Through this approach, pathological changes in tau protein can be targeted more precisely, offering new directions for the treatment of neurodegenerative diseases.

Tau protein, under physiological conditions, functions as a microtubule-associated protein essential for the stabilization and assembly of microtubules, which is crucial for axonal growth (Guo et al, 2017). Recent studies have also revealed the involvement of tau in synaptic functions, particularly highlighting its importance in promoting spine formation, dendritic elongation, and synaptic plasticity (Velazquez et al, 2018). Consequently, therapeutic strategies targeting tau protein might raise concerns about whether its clearance could disrupt its normal physiological functions. However, studies involving experimental animals have demonstrated that tau absence does not cause noticeable abnormalities (Biundo et al, 2018; Roberson et al, 2007). Depletion of tau does not affect axonal elongation in cultured neurons (Tint et al, 1998). These observations suggest that even in scenarios of reduced or absent tau protein, alternative microtubule-associated proteins may compensate and perform the functions of tau (Takei et al, 2000).

Excessive phosphorylation of tau protein is a critical factor in impaired synaptic plasticity, which is intimately linked to the severity of dementia in AD (Soria Lopez et al, 2019). Treatment with IsoLiPro significantly reduced dendritic loss in 3xTg-AD mice, a reduction likely attributable to the substantial reduction in hyperphosphorylated tau. Additionally, the post-treatment rise in synaptic functional markers, such as SYN-1 and PSD95, implies a rejuvenation of synaptic plasticity, crucial for enhancing cognitive functions in these mice. However, in the 5xFAD mice, which was subjected to a 12-week IsoLiPro treatment, significant cognitive improvements were not observed. This outcome could be due to the necessity for an extended treatment duration with IsoLiPro. Furthermore, the 5xFAD model presents predominantly amyloid plaque pathology rather than tau pathology, highlighting the significant role of tau in cognitive impairment (Iyaswamy et al, 2022; Wang et al, 2023). Targeted tau clearance in 3xTg-AD mice resulted in a more noticeable cognitive improvement, whereas in 5xFAD mice, where tau pathology is less pronounced, cognitive enhancement was not as evident as that observed in those 3xTg-AD mice.

Although IsoLiPro can promote tau degradation through inhibiting USP11, as a lithium-containing molecule, the degraded lithium from IsoliPro may also exert other lithium-related bioactivities, such as GSK-3β inhibition, which may contribute to the anti-amyloidopathy effects of IsoLiPro (Alvarez et al, 2002; Phiel et al, 2003). Interestingly, previous studies further indicated that USP11 inhibition may protect amyloid aggregation-induced paralysis in *C elegans* and increase BACE1 degradation in AD model mice (Basic et al, 2021; Wu et al, 2022), which may also contribute to the ameliorated amyloid deposition induced by IsoLiPro.

Neuroinflammation plays a pivotal role in the pathology of AD and influences the disease's progression. Astrocytes and microglia, as the primary immune effectors in the central nervous system

(CNS), are actively involved in the inflammatory response of AD (Bradburn et al, 2019). Under neuroinflammatory conditions, these glial cells are activated, releasing pro-inflammatory factors that accelerate the progression and pathology of AD (Chew and Petretto, 2019). In our study, we focused on the expression of Iba1 and GFAP, markers for microglia and astrocytes, respectively, in the brains of 5xFAD and 3xTg-AD mice. Treatment with IsoLiPro significantly reduced the expression of Iba1 and GFAP in 5xFAD mice. However, in the 3xTg-AD mice, where there is a lower prevalence of glial cells, particularly astrocytes, no significant changes were observed. This highlights the differences in glial cell expression between different animal models. The 5xFAD mice, predominantly affected by Aβ deposition, exhibit a more significant glial cell presence than the 3xTg-AD mice (Chiba et al, 2009). Furthermore, the variation in Aβ levels before and after IsoLiPro treatment, along with the deposition of glial cells around Aβ, also indirectly reflects the role of Aβ in activating glial cells. Ultimately, IsoLiPro mitigates the progression of AD by reducing Aβ deposition and curtailing the initiation of inflammatory responses.

However, microglia and astrocytes are important for maintaining neuronal function and microenvironment homeostasis via complicated cell-cell interactions (Salter and Stevens, 2017). Moreover, the dysregulated or altered microglia or astrocyte function will contribute to the pathogenesis and progression of neurodegenerative diseases, including AD (Ransohoff, 2016). From this end, further investigations of USP11 in those non-neuron cells, such as expression levels with or without IsoLiPro are important for comprehensive understanding of USP11 function and mechanisms underlying IsoLiPro therapy. Although in our present study, we did not investigate the specific expression of USP11 in non-neuronal glial cells, previous studies have indicated that some other USP members such as USP18 also express in microglia (Goldmann et al, 2015). Future studies are planned to delve deeper into the functional outcomes of USP11 inhibition in both neuronal and glial cells, to better understand the balance between potential therapeutic benefits and risks associated with targeting USP11 in different cell types within the central nervous system. This provides us a new perspective for future studies.

In conclusion, in our present study, we found that IsoLiPro can lower USP11 protein levels both in vivo and in vitro, mainly through modulating protein degradation without impacting directly the USP11 enzyme activity, and consequently ameliorate AD pathologies in transgenic AD animal model. These data highlight USP11 as a potential therapeutic target of AD by inhibiting its protein level and consequently enhancing the ubiquitination and degradation of tau protein. However, further explorations are still required to quantify IsoLiPro in the brain to directly reflect the exact dose-response relationships and detailed molecular mechanisms in vivo. Moreover, it is also important for us to clarify the direct interactions between IsoLiPro and USP11 in vivo in the brain to further support our hypothesis and make our conclusions more convincing.

## Study approval

All the animal experimental procedures were approved by the Animal Ethics Committee of Guizhou Medical University (SYXK (Gui) 2023-0002). Animals received humane care in compliance with the "Guiding Principles in the Care and Use of Animals" (China).

## The paper explained

### Problem

Alzheimer's Disease (AD) is the most common neurodegenerative disease globally, severely affecting cognitive function. While its exact cause is unclear, abnormal aggregation of misfolded β-amyloid and tau proteins is central to its progression. Recent studies highlight ubiquitin-specific protease 11 (USP11) as a key regulator of tau deubiquitination, promoting tau aggregation and AD pathology. Inhibiting USP11, by blocking its activity or reducing its protein levels, offers a potential therapeutic strategy. However, the absence of USP11-specific drugs emphasizes the urgent need for new small-molecule inhibitors targeting USP11 for AD treatment.

### Results

Our research introduces IsoLiPro, a novel lithium isobutyrate-L-proline coordination compound, which effectively reduces USP11 protein levels and enhances tau ubiquitination in vitro. Furthermore, long-term oral administration of IsoLiPro significantly alleviates tau pathologies in the brains of AD transgenic mice, as demonstrated by reductions in both total and phosphorylated tau levels. Additionally, IsoLiPro substantially decreases β-amyloid deposition and synaptic damage, leading to improved cognitive functions in these animal models.

### Impact

These findings suggest that IsoLiPro, as a novel small-molecule USP11 inhibitor, can effectively alleviate AD-like pathologies and improve cognitive functions. This highlights its potential as a multi-targeting therapeutic agent for AD, offering new treatment strategies.

## Data availability

This study includes no data deposited in external repositories.

The source data of this paper are collected in the following database record: biostudies:S-SCDT-10_1038-S44321-024-00146-7.

## Peer review information

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

## Acknowledgements

The authors would like to thank Dr. Ximei Wu, Dr. Xiangnan Zhang, and Mr. Yuedong Yu for their constructive comments and technical support. This study was supported by the High-level Talent Foundation of Guizhou Medical University and Hangzhou City University (YJ19017, HY2020, C22921 JT), National Natural Science Foundation of China (NSFC) (82171423 and 82060211, JT; 31530089, YS; 82460263, JC), Anyu Biopharmaceutics, Inc., Hangzhou (06202010204, JT), Strategic Priority Research Program of the Chinese Academy of Sciences (XDB39000000, YS), Scientific Research Project of higher education Institutions in Guizhou Province [192(2022), JC], Science and Technology Program of Guizhou Province [ZK(2023), General 301, JC], Guizhou Science and Technology Talent Cooperation Platform [CXTD[2023]003, XQ].

## Author contributions

**Yi Guo**: Data curation; Formal analysis; Investigation; Visualization; Methodology; Writing—original draft; Writing—review and editing. **Chuanbin Cai**: Resources; Data curation; Software; Formal analysis; Investigation; Visualization; Methodology. **Bingjie Zhang**: Data curation; Software; Formal analysis; Validation; Investigation; Methodology. **Bo Tan**: Investigation; Visualization. **Qinmin Tang**: Data curation; Formal analysis; Investigation. **Zhifeng Lei**: Validation; Investigation; Methodology. **Xiaolan Qi**: Data curation; Visualization; Methodology. **Jiang Chen**: Software; Investigation; Methodology; Project administration. **Xiaojiang Zheng**: Methodology; Project administration. **Dan Zi**: Project administration; Writing—review and editing. **Song Li**: Conceptualization; Resources; Supervision; Methodology; Writing—original draft; Project administration; Writing—review and editing. **Jun Tan**: Conceptualization; Data curation; Supervision; Funding acquisition; Project administration; Writing—review and editing.

Source data underlying figure panels in this paper may have individual authorship assigned. Where available, figure panel/source data authorship is listed in the following database record: biostudies:S-SCDT-10_1038-S44321-024-00146-7.

## Disclosure and competing interests statement

The authors declare no competing interests.

# Expanded View Figures

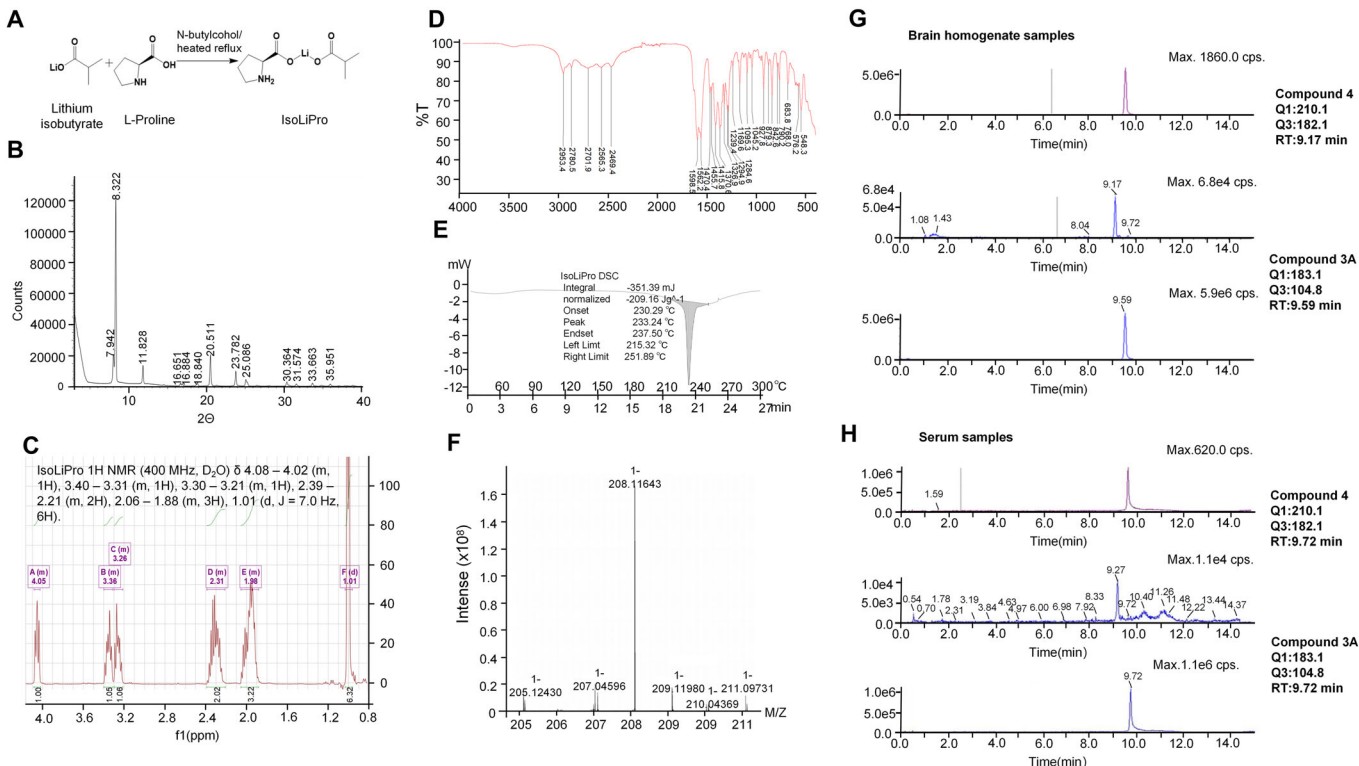

**Figure EV1.    Synthesis and characterization of IsoLiPro.**

(A) Reaction diagram - Reaction for IsoLiPro. (B) Powder X-ray diffraction analysis of n-butanol as a single solvent in a powder with IsoLiPro. (C) [1]H NMR spectrum of IsoLiPro in MeOD solvent. (D) Spectrum FT-IR of IsoLiPro. (E) Differential scanning calorimetry analysis of IsoLiPro. (F) Ion mass-to-charge ratios of IsoLiPro molecules tested with MRMS of Bruker solarix in negative ion mode. The detection of IsoLiPro (MW:210.1) whole molecule in brain homogenate (G) and serum (H) using tandem quadrupole mass spectrometry.

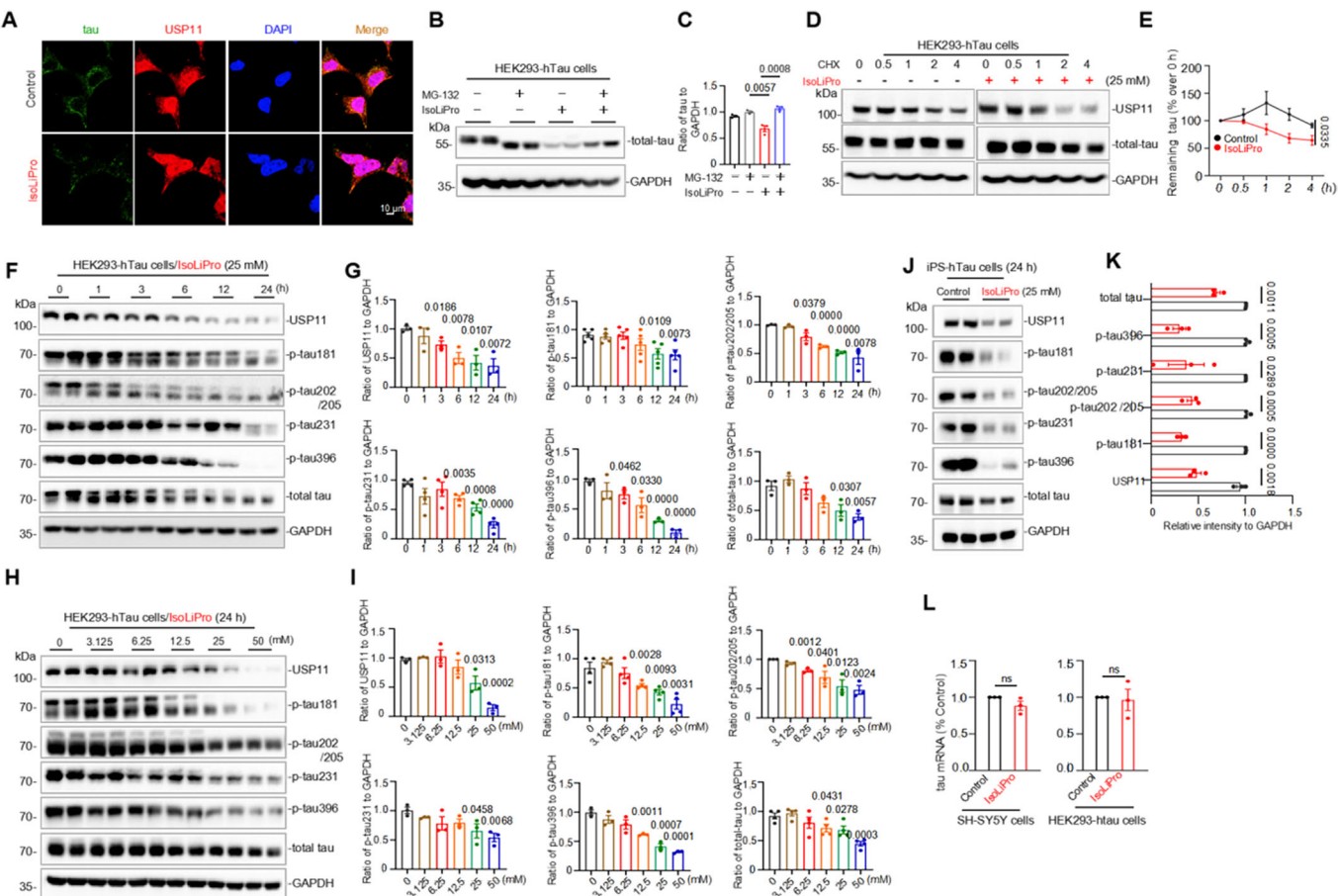

**Figure EV2. IsoLiPro markedly reduces levels of total and phosphorylated tau in cultured cells overexpressing wild-type full-length human tau.**

(A) IsoLiPro significantly reduces the co-localization of USP11 (red) and tau (green) in HEK293-hTau cells measured by immunofluorescence (IF). (B, C) IsoLiPro (25 mM for 24 h) markedly accelerates tau clearance, altered by inhibition of the proteasomal- (MG132, 10 μM) degradation signaling pathways in HEK293-hTau cells. Data were represented as means ± SEM ($n = 3$ samples per group). $P$ values were calculated using one-way ANOVA followed by Tukey's HSD test. (D, E) IsoLiPro markedly enhances tau turnover in HEK293-hTau cells (CHX, 100 μg/mL). Data were represented as means ± SEM ($n = 3$ samples per group). $P$ values were calculated using two-way ANOVA, with comparisons made against the Control group. (F–I) IsoLiPro markedly reduces phospho- and total-tau levels in a time- (F, G) and dose-dependent (H, I) manner in HEK293-hTau cells. Data were represented as means ± SEM ($n = 3$ to 5 samples per group). $P$ values were calculated using multiple $t$-tests, with comparisons made against the 0 h or 0 mM group. (J, K) IsoLiPro significantly inhibits phospho- and total-tau levels in iPS-hTau cells. Data were represented as means ± SEM ($n = 3$ samples per group). $P$ values were calculated using a two-tailed $t$-test, with comparisons made against the Control group. (L) IsoLiPro markedly lowers total tau protein levels without affecting mRNA levels in SH-SY5Y or HEK293-htau cells. Data were represented as means ± SEM ($n = 3$ samples per group). $P$ values were calculated using a two-tailed $t$-test, with comparisons made against the Control group. Source data are available online for this figure.

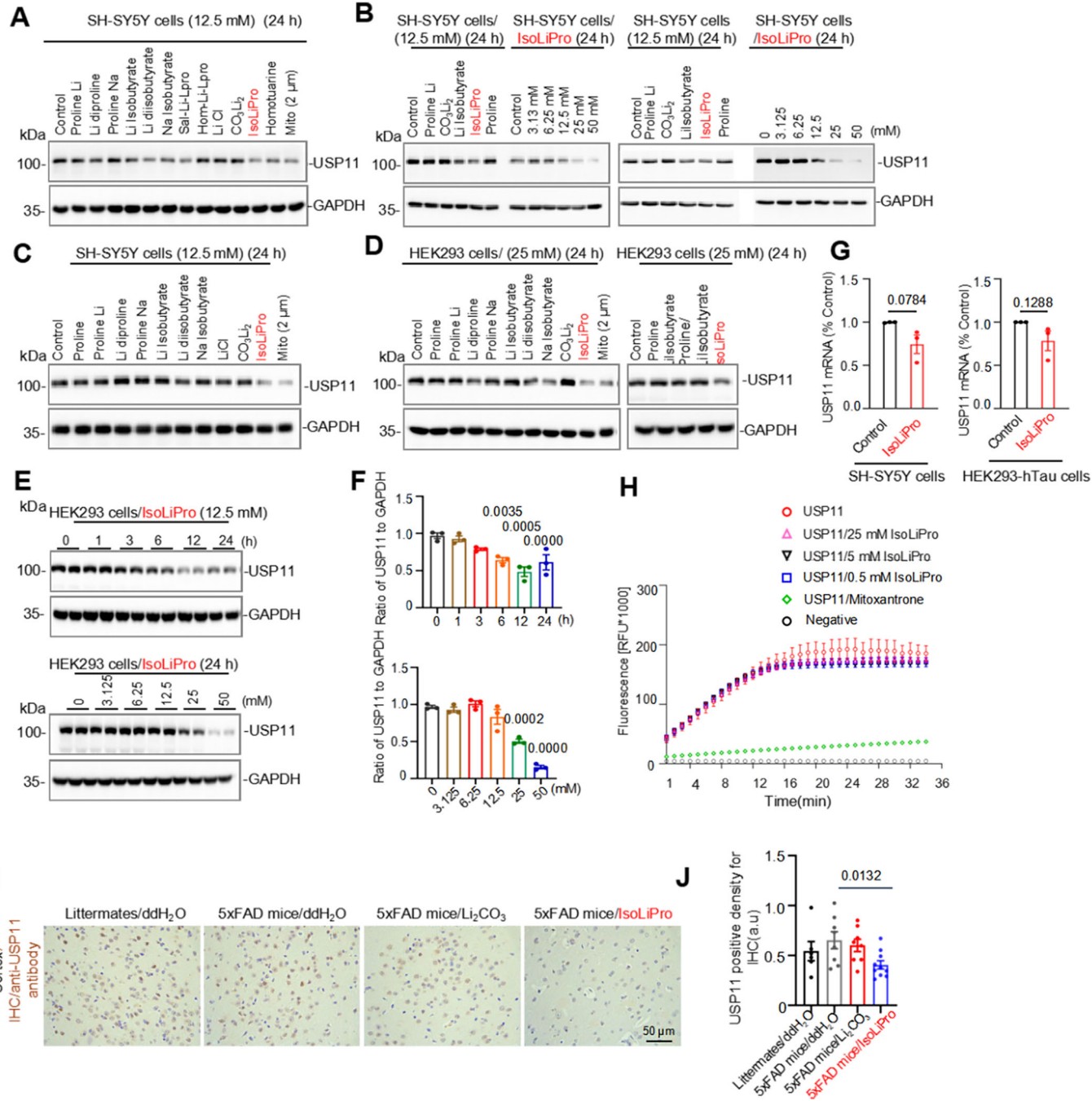

**Figure EV3. IsoLiPro markedly decreases USP11 protein levels in SH-SY5Y and HEK293 cells.**

(A–C) Representative blots depicting the alterations in USP11 protein levels following treatment with diverse small molecule drugs in SH-SY5Y cells. (D–F) Representative blots illustrate the changes in USP11 protein levels upon treatment with a range of small molecule drugs in HEK293 cells. Data were represented as means ± SEM ($n = 3$ samples per group). $P$ values were calculated using multiple $t$-tests, with comparisons made against the 0 h or 0 mM group. (G) USP11 mRNA level alteration after IsoLiPro treatment. Data were represented as means ± SEM ($n = 3$ samples per group). $P$ values were calculated using a two-tailed $t$-test, with comparisons made against the Control group. (H) Representative Ub-AMC cleavage assay of USP11. The fluorescence signal (RFU*103) is plotted against the time [min]. Data were represented as means ± SEM ($n = 3$ samples per group). (I, J) Immunohistochemical images revealing USP11 protein levels in the cortex of 5xFAD mice. Data were represented as means ± SEM ($n = 3$ to 5 mice per group, with two cortical sections analyzed from each mouse). $P$ values were calculated using multiple $t$-tests, with comparisons made against the 5xFAD mice/ddH$_2$O group. Source data are available online for this figure.

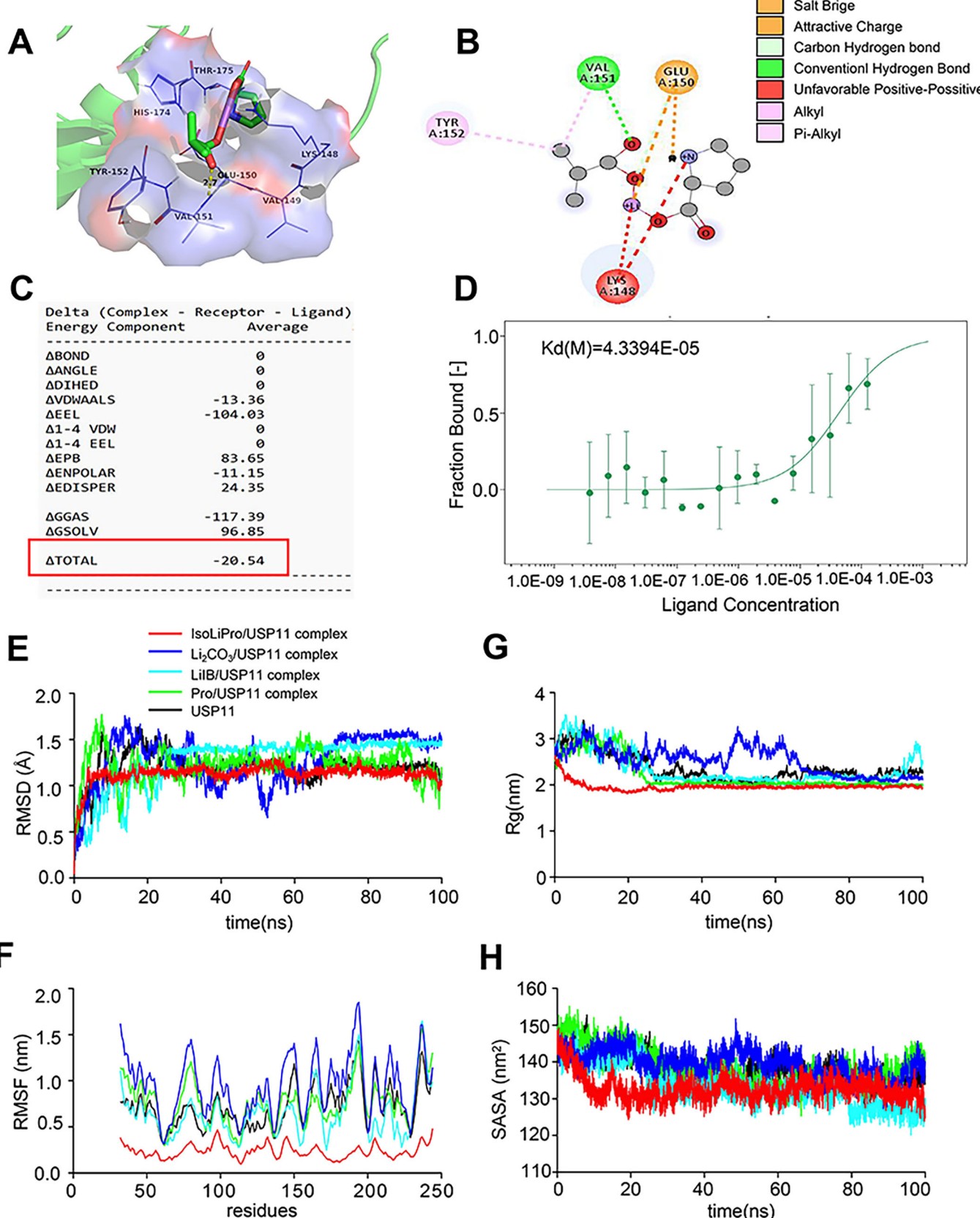

◀     **Figure EV4.   Binding affinity of IsoLiPro to USP11.**

(**A**) The 3D image illustrates IsoLiPro's binding to the USP11 protein. (**B**) The 2D image reveals IsoLiPro's interaction with several amino acids, including Lys148, Glu150, Val151, and Tyr152. (**C**) The binding free energy between IsoLiPro and USP11 was calculated using the MM-PBSA method. (**D**) Microscale thermophoresis (MST) results: IsoLiPro binding to USP11 protein yielded a dissociation (Kd) of 4.3394E-05 M. Data were represented as means ± SEM ($n = 3$ samples per group). (**E–H**) Analysis of root mean square deviation (RMSD) (**E**), root mean square fluctuation (RMSF) (**F**), radius of gyration (Rg) (**G**), and solvent accessible surface area (SASA) (**H**) of USP11 and its complexes with IsoLiPro, LiIB, $Li_2CO_3$, Pro during 100 ns MD simulations. Source data are available online for this figure.

                                                                        

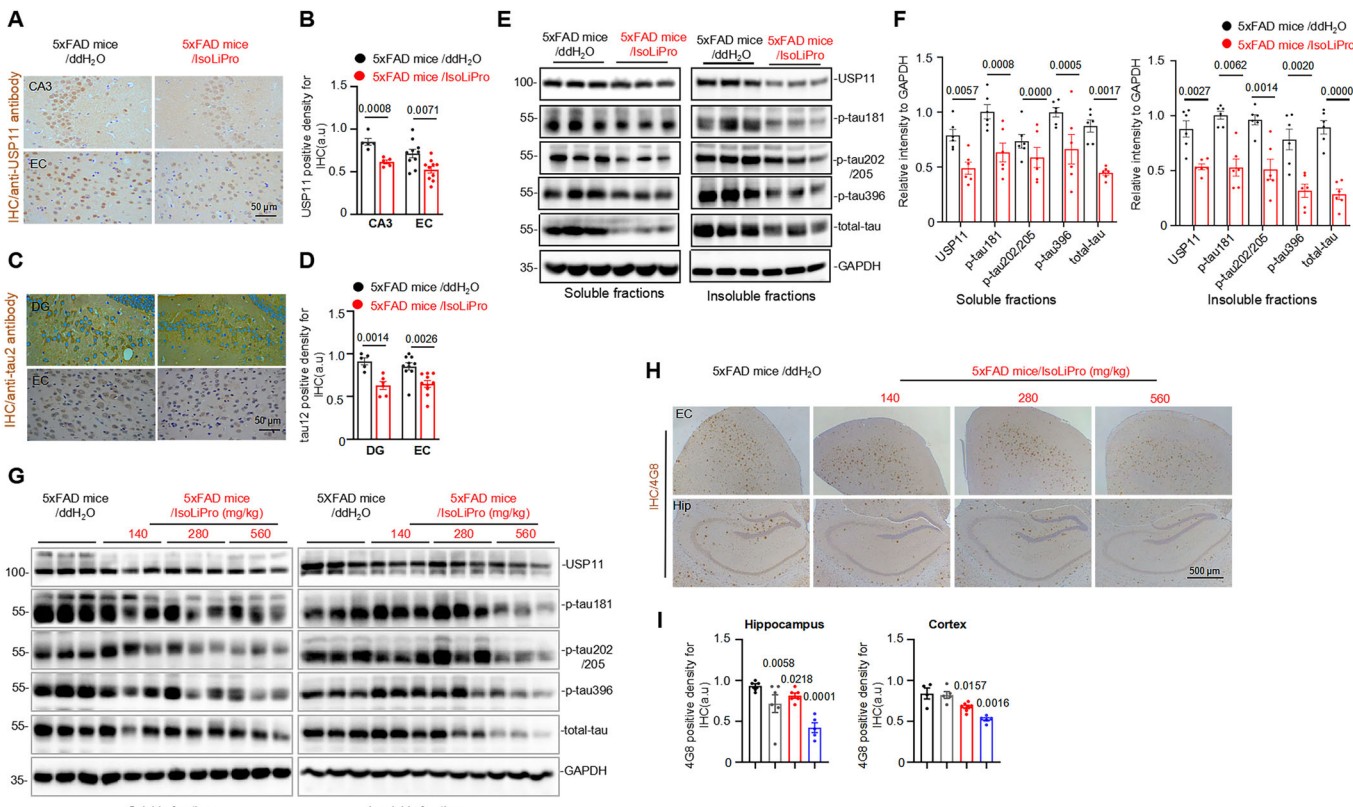

**Figure EV5. IsoLiPro markedly induces tau clearance in the brain of 5xFAD mice.**

(A, B) In 10-week-old 5xFAD mice, a 12-week oral administration of IsoLiPro (560 mg/kg) significantly decreased USP11 levels in both the hippocampus and cortex, as assessed by IHC. Data were represented as means ± SEM ($n = 5$ mice per group, with two cortical sections analyzed from each mouse). $P$ values were calculated using a two-tailed $t$-test, with comparisons made against the 5xFAD mice/ddH$_2$O group. (C, D) Oral gavage IsoLiPro notably decreased the levels of total-tau in the hippocampus and cortex of the 5xFAD mice measured by IHC. Data were represented as means ± SEM ($n = 5$ mice per group, with two cortical sections analyzed from each mouse). $P$ values were calculated using a two-tailed $t$-test, with comparisons made against the 5xFAD mice/ddH$_2$O group. (E, F) Oral gavage IsoLiPro significantly decreased total tau and the tau phosphorylated at multiple-doses AD-related sites (soluble and insoluble fractions) measured by Western blotting. Data were represented as means ± SEM ($n = 6$ mice per group). $P$ values were calculated using a two-tailed $t$-test, with comparisons made against the 5xFAD mice/ddH$_2$O group. (G) Western blot analysis was utilized to validate the effect of orally administered IsoLiPro at varying concentrations on the expression levels of total tau and the tau phosphorylated at multiple-doses AD-related sites (soluble and insoluble fractions). (H, I) Representative IHC images illustrate alterations in 4G8-positive amyloid plaques in 5xFAD mice subsequent to treatment with different concentrations of IsoLiPro. Data are represented as means ± SEM ($n = 3$ to 5 mice per group). Some samples from the 5xFAD mice/ddH2O and 5xFAD mice/560 mg/kg groups were re-used from Fig. 5E. $P$ values were calculated using multiple $t$-tests, with comparisons made against the 5xFAD mice/ddH$_2$O. Source data are available online for this figure.

