## [Peer Review File · EMBO Molecular Medicine]

Targeting USP11 regulation improves neuropathologies and cognition in Alzheimer transgenic mice

Yi Guo, Chuanbin Cai, Bingjie Zhang, Bo Tan, Qinmin Tang, Zhifeng Lei, Xiaolan Qi, Jiang Chen, Xiaojiang Zheng, Dan Zi, Song Li, and Jun Tan

Corresponding author: Song Li (lisong@dmu.edu.cn) , Jun Tan (tanjun@anyuhz.cn)

Review Timeline:

Submission Date:	30th May 24
Editorial Decision:	21st Jun 24
Revision Received:	8th Aug 24
Editorial Decision:	3rd Sep 24
Revision Received:	13th Sep 24
Accepted:	16th Sep 24

Editor: Jingyi Hou

Transaction Report:

21st Jun 2024

Dear Dr. Li,

Thank you again for submitting your work to EMBO Molecular Medicine. We have now heard back from the three referees who evaluated your manuscript. As you will see from the reports below, the referees find the topic of your study of potential interest. However, they raise substantial concerns on your work, which should be convincingly addressed in a major revision of the present manuscript.

The referees' recommendations are relatively straightforward, so there is no need for me to reiterate the points listed below. All the issues raised by the reviewers need to be carefully addressed.

We would welcome the submission of a revised version within three months for further consideration. Please feel free to contact me in case you would like to discuss in further detail any of the issues raised by the referees.

As you may already know, our editorial policy allows in principle a single round of major revision, and it is therefore essential to provide responses to the reviewers' comments that are as complete as possible.

I look forward to receiving your revised manuscript soon.

Yours sincerely,
Jingyi

Jingyi Hou
Editor
EMBO Molecular Medicine

We require:

- 1) A .docx formatted version of the manuscript text (including legends for main figures, EV figures and tables). Please make sure that the changes are highlighted to be clearly visible.
- 2) Individual production quality figure files as .eps, .tif, .jpg (one file per figure). For guidance, download the 'Figure Guide PDF': (<https://www.embopress.org/page/journal/17574684/authorguide#figureformat>).
- 3) A .docx formatted letter INCLUDING the reviewers' reports and your detailed point-by-point responses to their comments. As part of the EMBO Press transparent editorial process, the point-by-point response is part of the Review Process File (RPF), which will be published alongside your paper.
- 4) A complete author checklist, which you can download from our author guidelines (<https://www.embopress.org/page/journal/17574684/authorguide#submissionofrevisions>). Please insert information in the checklist that is also reflected in the manuscript. The completed author checklist will also be part of the RPF.

6) It is mandatory to include a 'Data Availability' section after the Materials and Methods. Before submitting your revision, primary datasets produced in this study need to be deposited in an appropriate public database, and the accession numbers and database listed under 'Data Availability'. Please remember to provide a reviewer password if the datasets are not yet public (see <https://www.embopress.org/page/journal/17574684/authorguide#dataavailability>).

13) Author contributions: You will be asked to provide CRediT (Contributor Role Taxonomy) terms in the submission system. These replace a narrative author contribution section in the manuscript.

14) A Conflict of Interest statement should be provided in the main text.

15) Every published paper now includes a 'Synopsis' to further enhance discoverability. Synopses are displayed on the journal webpage and are freely accessible to all readers. They include a short stand first (maximum of 300 characters, including space) as well as 2-5 one-sentence bullet points that summarize the paper. Please write the bullet points to summarize the key NEW findings. They should be designed to be complementary to the abstract - i.e. not repeat the same text. We encourage inclusion of key acronyms and quantitative information (maximum of 30 words / bullet point). Please use the passive voice. Please attach these in a separate file or send them by email, we will incorporate them accordingly.

Please also suggest a striking image or visual abstract to illustrate your article as a PNG file 550 px wide x 300-800 px high.

***** Reviewer's comments *****

Referee #1 (Comments on Novelty/Model System for Author):

Lack of characterization of the compound engaging the target in vivo and it remains unclear what dose of the compound reaches the brain inhibiting USP11

Referee #1 (Remarks for Author):

In the current study, the authors showed that IsoLiPro, a unique lithium isobutyrate-L-proline coordination compound, lowers USP11 protein level and enhances tau ubiquitination in vitro. Long-term oral administration of IsoLiPro dramatically alleviates tau pathologies in the brain of AD transgenic mice, as evidenced by the reduced both total and phosphorylated tau. Moreover, IsoLiPro also significantly lessens β -amyloid deposition and synaptic damage, improving cognitive functions in these models. Overall, the study is interesting. However, the data are not convincing. The present data are not sufficient to support the main claim.

Referee #2 (Comments on Novelty/Model System for Author):

In this manuscript, the authors synthesized a new compound (IsoLiPro), which docks and binds to USP11 with K_d of ~43 micromolar. They show that low millimolar concentrations of IsoLiPro decrease USP11 protein in cultured cells and mouse brains, resulting in reduced tau and phospho-tau likely through increasing tau ubiquitination. IsoLiPro does not appear to affect the level of USP25, another DUB implicated in AD pathology. Oral administration of IsoLiPro (560mg/kg) for 16 weeks also reduces tau, phospho-tau, and neuroinflammation in 3xTG and 5xFAD mice. Interestingly, IsoLiPro also appears to decrease Abeta deposition in 5xFAD mice. Finally, they show that IsoLiPro ameliorates synaptic and spatial memory deficits in 3xTg and 5xFAD mice.

This is an interesting paper that highlights the DUB USP11 as a molecular target to ameliorate tau pathogenesis, aligning with the previously documented role of USP11 in promoting tau pathology. The results are potentially therapeutically relevant and significant. However, the justification for synthesizing IsoLiPro to target USP11 is entirely missing or unclear, several key experiments are missing, and scientific rigor of the study requires significant improvement. The following critiques should be addressed.

1. Clear justification of why IsoLiPro was synthesized to target USP11 needs to be clearly stated. Why not other USPs or any other AD-related target?
2. Molecular docking is done for USP11. Docking should be done for other USPs such as USP25, the latter level which is unaffected by IsoLiPro.
3. Microscale thermophoresis experiments indicate IsoLiPro binding to USP11 at K_d of ~43 micromolar. However, there are no experiments showing that IsoLiPro directly inhibits USP11 DUB activity, for example, using recombinant USP11.
4. While the K_d of IsoLiPro-USP11 binding is in the micromolar range, all cell-based experiments use IsoLiPro at the millimolar range, indicating that IsoLiPro has relatively weak USP11 inhibitory activity. What is the evidence that USP11 is a significant target of IsoLiPro at such high concentration. In other words, does IsoLiPro reduce tau levels in the absence of USP11?
5. There is no quantification of many experiments and figures, which weakens the scientific rigor of the manuscript. Quantification should be provided for the following figures: 1E, 4A-C, 5A-C, 6A-C.
6. Fig. 1D appears to show that IsoLiPro increases USP11, not decreases it. This contrasts with IHC and cell-based data. Please explain.
7. Fig. 2E shows tau increasing after CHX in the control condition. Please explain.

8. Fig. 3D requires labels on the Western blot.
9. How do the authors explain the decrease in Aβ deposition after IsoLiPro treatment in 5xFAD mice? This should be mentioned in the discussion.
10. Oral administration uses a very high dose of IsoLiPro for a prolonged period of 16 weeks. At such high concentration, there could be toxicity in multiple organs. Given that IsoLiPro is a new compound, in vivo toxicity should be evaluated in all major organs. Further, the pharmacokinetic profile of IsoLiPro should be determined.
11. English writing and grammar should be edited by a native English speaker.

Referee #3 (Comments on Novelty/Model System for Author):

Li₂CO₃ and LiIB have not been established as inhibitors of USP11, and their binding affinity to the enzyme necessitates validation through the use of a positive control, such as mit, in the second phase of the results section.

Referee #3 (Remarks for Author):

This study presents the design of a novel small molecule compound, specifically engineered to target USP11, a pivotal player in Alzheimer's Disease (AD) pathogenesis. By functioning as an upstream regulator of tau protein dynamics, the compound efficiently suppresses USP11 activity, resulting in a dual effect: reduced tau protein phosphorylation and enhanced K48-linked ubiquitination. This approach holds promise for potential therapeutic intervention in AD by targeting a key molecular event in the disease progression. However, there are still some problems in this study.

1. The selection of lithium isobutyrate as a substrate for compound design warrants further elucidation. It is crucial to explore the rationale behind this choice and highlight any potential advantages that this design strategy may offer in terms of its biochemical properties and therapeutic implications.
2. Li₂CO₃ and LiIB have not been established as inhibitors of USP11, and their binding affinity to the enzyme necessitates validation through the use of a positive control, such as mit, in the second phase of the results section.
3. The compound demonstrated its ability to cross the blood-brain barrier, as discussed in the methodology, but the results section did not explicitly illustrate the brain penetration following oral administration.
4. The authors did not specifically refer to the sample size of the statistic, e.g. some of the results of the WB, although significant, did not have duplicate samples and were not statistically significant, and it was necessary to ensure that there were at least three independent duplicate samples for each set of results
5. As a novel drug, in vitro cellular administration needs to be addressed whether the drug is damaging to the cells, thus, cellular activity assay is a requirement.
6. Delivery of drugs from the periphery requires verification of biocompatibility, thus ensuring that the new drug is not harmful to other organs.
7. The study demonstrates that the novel compounds influence tau proteasomal degradation, leading to a decrease in tau protein levels. However, it is important to note that the established USP11 inhibitor, Mitoxantrone, impacts USP11 mRNA expression, subsequently affecting USP11 activity. The current article does not address whether the new compounds similarly affect USP11 proteasomal degradation and, consequently, tau protein levels.
8. Although Alzheimer's disease (AD) is a chronic condition, the authors' proposed 16-week administration duration raises the question of whether an alternative delivery method, such as intracranial cannula, could be explored for a potentially shorter and more targeted administration of the drug.
9. The expression of USP11 in neurons specifically should be investigated, as its inhibition in non-neuronal glial cells could potentially lead to a decrease in neuronal damage and cognitive function impairment.

***** Response to reviewer's comments *****

Referee #1 (Comments on Novelty/Model System for Author):

Lack of characterization of the compound engaging the target *in vivo* and it remains unclear what dose of the compound reach the brain inhibiting USP11

Response: Thank you very much for your comments. In our present study, we found that IsoLiPro, a coordination compound composed of lithium isobutyrate and L-proline, significantly decrease the protein level of USP11, which may contribute to its ameliorating effects on AD pathologies. In addition, under therapeutic dosage, no significant side effects and pathological changes in tissues were observed after chronic IsoLiPro exposure *in vivo*, together with no significant cytotoxicity *in vitro* (**Appendix Fig. S6**). These toxicological data suggest a good safety profile of IsoLiPro, comparing with the previously reported serious cardiotoxicity and the increased risk of myeloid leukemia induced by the well-known USP11 inhibitor, mitoxantrone.

Moreover, it is hypothesized that the intact whole molecule of IsoLiPro rather than the decomposed lithium ion is the real molecule responsible for the decreased USP11 level induced by IsoLiPro, because in our present study lithium carbonate shows no impacts on USP11 level as IsoLiPro dose.

By using Tandem quadrupole mass spectrometry (QTRAP® 5500 system), we have successfully established a method to detect the whole molecule IsoLiPro in serum and brain homogenates from those mice treated by IsoLiPro (i.g., 560 mg/kg). Our analysis clearly identified a compound 4 with molecular weight of 210.1 (**Fig. EV1G and EV1H** in the revised manuscript), suggesting the existence of whole molecule IsoLiPro (calculated molecular weight 209.17, C₉H₁₆LiNO₄) and its penetration through blood brain barrier to reach into CNS to exert biological effects.

Also, oral administration of IsoLiPro significantly decreased USP11 level in the brain, consistent with the notable inhibition effect observed in cell-based systems *in vitro*. This lowered USP11 level was not observed in cells treated with lithium or

isobutyric acid alone, further suggesting that IsoLiPro likely decreases USP11 level in its whole molecule form.

Unfortunately, the quantification of whole molecule IsoLiPro reaching into brain tissues after system administration is still the biggest challenge of our present study. Currently, we use total lithium content, composed of both whole molecule IsoLiPro and degraded lithium ion, as the concentration of IsoLiPro reaching into the brain. We are now engaging to develop reliable methods for the quantification of IsoLiPro whole molecule, so as to evaluate its real dose-response relationship and exact molecular mechanisms.

Fig. EV1 (G-H) The detection of IsoLiPro whole molecule in brain homogenate (G) and serum (H) using Tandem quadrupole mass spectrometry.

Appendix Fig. S6 Toxicity evaluation of IsoLiPro. (A) Possible pathological changes in the heart, liver, spleen, lung and kidney of mice administered IsoLiPro (560 mg/kg, gavage) for 16 weeks were evaluated by HE staining. (B) Monitoring body weights of mice during the course

of treatments. (C) SH-SY5Y and HEK293 cells viability was detected by CCK-8 assay after IsoLiPro exposure (up to 50 mM) for 24 hours.

Referee #1 (Remarks for Author):

In the current study, the authors showed that IsoLiPro, a unique lithium isobutyrate-L-proline coordination compound, lowers USP11 protein level and enhances tau ubiquitination in vitro. Long-term oral administration of IsoLiPro dramatically alleviates tau pathologies in the brain of AD transgenic mice, as evidenced by the reduced both total and phosphorylated tau. Moreover, IsoLiPro also significantly lessens β -amyloid deposition and synaptic damage, improving cognitive functions in these models. Overall, the study is interesting. However, the data are not convincing. The present data are not sufficient to support the main claim.

Response: Thank you very much for your positive evaluations. Although it is still challenging currently for us to establish methods to stably quantify IsoLiPro whole molecule in the brain, this coordination compound can be detected qualitatively in both the brain and the serum of mice treated with single dose of IsoLiPro by gavage. Much more importantly, the identification of whole molecule IsoLiPro in the brain is pivotal for us to speculate the exact molecular-basis of its USP11 inhibition activity and further explore the underlying molecular mechanisms. In our revised manuscript, we presented the qualitative data of Tandem quadrupole mass spectrometry assay to confirm the existence of IsoLiPro whole molecule (**Fig. EV1G and EV1H**).

Also, oral administration of IsoLiPro significantly decreased USP11 level in the brain, consistent with the notable inhibition observed in cell-based systems in vitro. This lowered USP11 level was not observed in cells treated with lithium or isobutyric acid alone, further suggesting that IsoLiPro likely decreases USP11 level in its whole molecule form. In addition, the data of USP11 activity assay indicated that IsoLiPro failed to directly suppress USP11 activity (**Fig. EV3H**), suggesting the involvement of protein degradation in IsoLiPro-induced USP11 downregulation.

Taken together, in our present study, we found that IsoLiPro can lower USP11 protein level both *in vivo* and *in vitro*, mainly through modulating protein degradation without impacting directly the USP11 enzyme activity, and thus ameliorate AD pathologies in animal model. These data highlight USP11 as potential therapeutic target of AD. In addition, the activity and safety profiles are different from mitoxantrone, suggesting a promising potential of IsoLiPro in future clinical translation. However, further explorations are still required to quantify IsoLiPro in the brain to directly reflect the exact dose-response relationships and detailed molecular mechanisms *in vivo*. Moreover, it is also important for us to clarify the direct interactions between IsoLiPro and USP11 *in vivo* in the brain to further support our hypothesis and make our conclusions more convincing. These points have been added in our revised manuscript (page 14-15, lines 352-356; page 18, lines 435-436; page 26-27, lines 644-654; pages 32, lines 777-787).

Referee #2

In this manuscript, the authors synthesized a new compound (IsoLiPro), which docks and binds to USP11 with K_d of ~43 micromolar. They show that low millimolar concentrations of IsoLiPro decreases USP11 protein in cultured cells and mouse brains, resulting in reduced tau and phospho-tau likely through increasing tau ubiquitination. IsoLiPro does not appear to affect the level of USP25, another DUB implicated in AD pathology. Oral administration of IsoLiPro (560mg/kg) for 16 weeks also reduces tau, phospho-tau, and neuroinflammation in 3xTG and 5xFAD mice. Interestingly, IsoLiPro also appears to decrease Abeta deposition in 5xFAD mice. Finally, they show that IsoLiPro ameliorates synaptic and spatial memory deficits in 3xTg and 5xFAD mice.

This is an interesting paper that highlights the DUB USP11 as a molecular target to ameliorate tau pathogenesis, aligning with the previously documented role of USP11 in promoting tau pathology. The results are potentially therapeutically relevant and significant. However, the justification for synthesizing IsoLiPro to target USP11 is

entirely missing or unclear, several key experiments are missing, and scientific rigor of the study requires significant improvement. The following critiques should be addressed.

Response: Thank you very much for your positive evaluations and constructive comments on our manuscript. We have responded point by point and revised our manuscript accordingly.

1. Clear justification of why IsoLiPro was synthesized to target USP11 needs to be clearly stated. Why not other USPs or any other AD-related target?

Response: IsoLiPro was originally designed and synthesized as a novel organic lithium compound and therapeutic candidate for AD, considering the previously reported multi-targeting benefits of lithium on cognition and AD pathologies, including the decreased A β production and tau phosphorylation. Interestingly, our preliminary data further found that, despite the A β and p-tau ameliorations, IsoLiPro also significantly decreased total tau level, which was distinct from those inorganic lithium containing compounds. Due to the recently reported pivotal roles of USP11 in tau pathology in AD (*Yan et al., Cell, 2022*), we further determined the possible docking of IsoLiPro to USP11 and its inhibiting effects on USP11 activity and protein level. Fortunately, our data indicated that IsoLiPro decreases USP11 protein level and enhances tau ubiquitination level, which might contribute to the downregulated total tau. Further experiments are still required for exploring the exact mechanisms underlying the USP11 inhibition of IsoLiPro.

In addition, it has been reported that a total of 22 USP family members are expressed in CNS. Among these proteins, USP11, USP13 and USP14 have been identified to correlated with tau degradation (*Liu et al., J Alzheimers Dis, 2019*; *Boselli et al., J Biol Chem, 2017*; *Yan et al., Cell, 2022*). USP14 knockdown has been shown to marginally reduced tau level (*Yan et al., Cell, 2022*), and USP14 aptamers enhances tau degradation through the proteasome (*Lee et al., Sci Rep, 2015*). In contrast, phospho-tau level has been reported to be increased in USP14^{-/-} mice (*Jin et al., PloS one, 2012*), and that USP14 inhibition impairs autophagy (*Kim et al., Cell*

Rep, 2018). All these inconsistencies excluded USP14 for further considerations. As for the USP13, knocking it down alongside USP11 did not result in a significant additional reduction in tau levels, suggesting functional redundancy (Yan et al., Cell, 2022). Considering the more abundant expression level of USP11 in CNS (Yan et al., Cell, 2022) and the higher binding affinity of IsoLiPro to USP11 than USP13 (our supplementary data, **Appendix Fig. S2F**), we finally selected USP11 as the primary target for our present study.

USP siRNA screen identifies USP11 as positive tau regulators (data from Yan et al., 2022 Cell).

Appendix Fig. S2 Binding affinity of IsoLiPro to USP11 (-4,57 kcal/mol, **D**) and USP13 (-2.86 kcal/mol, **F**).

References

- Yan, Y., Wang, X., Chaput, D., Shin, M.K., Koh, Y., Gan, L., Pieper, A.A., Woo, J.A., and Kang, D.E. (2022). X-linked ubiquitin-specific peptidase 11 increases tauopathy vulnerability in women. *Cell* 185, 3913-3930.e3919.
- Liu, X., Hebron, M.L., Mulki, S., Wang, C., Lekah, E., Ferrante, D., Shi, W., Kurd-Misto, B., and Moussa, C. (2019). Ubiquitin Specific Protease 13 Regulates Tau Accumulation and Clearance in Models of Alzheimer's Disease. *J Alzheimers Dis* 72, 425-441.
- Boselli, M., Lee, B.H., Robert, J., Prado, M.A., Min, S.W., Cheng, C., Silva, M.C., Seong, C., Elsasser, S., Hatle, K.M., Gahman, T.C., et al. (2017). An inhibitor of the proteasomal deubiquitinating enzyme USP14 induces tau elimination in cultured neurons. *J Biol Chem* 292, 19209-19225.
- Lee, J.H., Shin, S.K., Jiang, Y., Choi, W.H., Hong, C., Kim, D.E., and Lee, M.J. (2015). Facilitated Tau Degradation by USP14 Aptamers via Enhanced Proteasome Activity. *Sci Rep* 5, 10757.
- Jin, Y.N., Chen, P.C., Watson, J.A., Walters, B.J., Phillips, S.E., Green, K., Schmidt, R., Wilson, J.A., Johnson, G.V., Roberson, E.D., Dobrunz, L.E., et al. (2012). Usp14 deficiency increases tau phosphorylation without altering tau degradation or causing tau-dependent deficits. *PLoS One* 7, e47884.
- Kim, E., Park, S., Lee, J.H., Mun, J.Y., Choi, W.H., Yun, Y., Lee, J., Kim, J.H., Kang, M.J., and Lee, M.J. (2018). Dual Function of USP14 Deubiquitinase in Cellular Proteasomal Activity and Autophagic Flux. *Cell Rep* 24, 732-743.

2. Molecular docking is done for USP11. Docking should be done for other USPs such as USP25, the latter level which is unaffected by IsoLiPro.

Response: This constructive suggestion is greatly appreciated. Accordingly, we performed molecular docking to predict the possible interactions of IsoLiPro with USP13 and USP25. Our data clearly indicated relatively lower binding affinities of IsoLiPro to USP13 (-2.86 kcal/mol) and USP25 (-4.41 kcal/mol) than USP11 (-4.57 kcal/mol) (**Appendix Fig. S2D-S2G**). Consistently, no alteration of USP25 protein level was observed after IsoLiPro exposure (**Fig. 1D and 1E**).

Appendix Fig. S2 (F-G) Binding affinity of IsoLiPro to USP13 (F) and USP25 (G).

Fig. 1 (D-E) Comparative analysis reveals IsoLiPro's selective inhibition of USP11 expression over USP25.

3. Microscale thermophoresis experiments indicate IsoLiPro binding to USP11 at Kd of ~43 micromolar. However, there are no experiments showing that IsoLiPro directly inhibits USP11 DUB activity, for example, using recombinant USP11.

Response: This suggestion is greatly appreciated. Additional experiment was carried out to determine the direct inhibiting activity of IsoLiPro on USP11 activity. The data was presented in **Fig. EV3H**. Unexpectedly, in contrast to positive control mitoxantrone, while IsoLiPro can bind to USP11 and reduce USP11 protein level, this compound failed to directly affect USP11 catalytic activity in vitro, as assessed by enzymatic activity assay. Our results suggest that IsoLiPro may impact USP11 function primarily through regulating protein stability, metabolism, post-translational modifications, or protein-protein interactions, rather than directly inhibiting its enzymatic activity. One example of compounds with these properties is small molecule that selectively degrade proteins of interest, which has emerged as

alternatives to selective chemical inhibitors. These degraders can be used as therapeutic modalities. Further studies are required to elucidate the specific mechanisms by which IsoLiPro affects USP11 level.

Fig. EV3 (H) Representative Ub-AMC cleavage assay of USP11. The fluorescence signal (RFU*10³) is plotted against the time [min].

4. While the K_d of IsoLiPro-USP11 binding is in the micromolar range, all cell-based experiments use IsoLiPro at the millimolar range, indicating that IsoLiPro has relatively weak USP11 inhibitory activity. What is the evidence that USP11 is a significant target of IsoLiPro at such high concentration. In other words, does IsoLiPro reduce tau levels in the absence of USP11?

Response: As we mentioned above in the responses to reviewer #1, it is hypothesized that the whole molecule of IsoLiPro may be responsible for the lowered USP11 level, rather than the lithium ion or isobutyrate or proline. Although IsoLiPro-USP11 binding is in the micromolar range in MST assay, some of the IsoLiPro molecules was degraded after administration in cell culture medium *in vitro* or after gavage in gastrointestinal and blood *in vivo*. Therefore, much higher dosage of IsoLiPro is required to reach effective concentration of whole molecule IsoLiPro to decrease USP11 protein levels for either cell-based or animal experiments. Moreover, while previous studies have reported the multi-targeting benefits of lithium on cognition and AD pathologies, including the ameliorated A β production and tau phosphorylation, no alteration of total tau have been reported, which is distinct from IsoLiPro. Therefore, the USP11 should be the important target of IsoLiPro, especially its whole molecule

form. It is estimated that IsoLiPro very likely failed to impact total tau in the absence of USP11.

5. There is no quantification of many experiments and figures, which weakens the scientific rigor of the manuscript. Quantification should be provided for the following figures: 1E, 4A-C, 5A-C, 6A-C.

Response: Thank you very much for this helpful suggestion. We have conducted necessary quantifications and statistical analysis to strengthen the scientific rigor of the revised manuscript. The detailed results can be found in **Fig. 1H** on page 43; **Fig. 4B, 4D, and 4F** on page 46; **Fig. 5C, 5D, and 5F** on page 47; **Fig. 6B, and 6D** on page 48.

Fig. 1H

Fig. 4B

Fig. 4D

Fig. 4F

Fig. 5C

Fig. 5D

Fig. 5F

Fig. 5F

Fig. 6B

Fig. 6D

6. Fig. 1D appears to show that IsoLiPro increases USP11, not decreases it. This contrasts with IHC and cell-based data. Please explain.

Response: Sorry for this misleading. The proteins bands are incorrectly labeled. Proper corrections have been made in the revised manuscript. The detailed results can be found in **Fig. 1F** on page 43.

Fig. 1 (F) Oral administration of IsoLiPro in 9.5-month-old 3xTg-AD mice for 16 weeks substantially reduced USP11 protein level in both soluble and insoluble fractions, as evidenced by Western blotting.

7. Fig. 2E shows tau increasing after CHX in the control condition. Please explain.

Response: Thank you very much for your careful review. As a ribosome inhibitor, CHX can restrict the translation elongation of eukaryotic protein synthesis. Once protein synthesis is inhibited by CHX, the level of intracellular proteins level will be regulated mainly through degradation. There can be great variation among proteins, especially during the early phase of chasing, which may result in a tendency of tau

increase (without statistical significance). It is normally recommended to start with 4-hour interval and chase till 24 hours. In our present study, we hypothesized that IsoLiPro can promote tau protein degradation through USP11 level inhibition. The decreased tau level induced by IsoLiPro may overlap which induced by CHX. Therefore, we detected the alterations of protein level in early 4 hours, during this period, we clearly observed a rapid decrease of tau protein level, indicating the degradation induced by IsoLiPro.

8. Fig. 3D requires labels on the Western blot.

Response: Your careful review is greatly appreciated. Proper corrections have been made in our revised manuscript (**Fig. 3D** on page 45).

Fig. 3 (D) Representative blots showing IsoLiPro inhibits the deubiquitination of CHIP-ubiquitinated recombinant tau by recombinant USP11 protein.

9. How do the authors explain the decrease in Abeta deposition after IsoLiPro treatment in 5xFAD mice? This should be mentioned in the discussion.

Response: This constructive comment is greatly appreciated. Although IsoLiPro can promote tau degradation through inhibiting USP11, as a lithium-containing molecule, the degraded lithium from IsoLiPro may also exert other lithium-related bioactivities, such as GSK-3 β inhibition, which may contribute to the anti-amyloidopathy effects of IsoLiPro (*Phiel et al., Nature, 2003; Alvarez et al., Bipolar disorder, 2002*). Interestingly, previous studies further indicated that USP11

inhibition may protect amyloid aggregation-induced paralysis in *C elegans* and increase BACE1 degradation in AD model mice, which may also contribute to the ameliorated amyloid deposition (*Basic et al.; JBC 2021; Wu et al., Health Research, 2022*). Proper contents have been added in the discussion section in our revised manuscript (page 30, lines 739-745).

References

- Phiel CJ, Wilson CA, Lee VM, Klein PS. GSK-3alpha regulates production of Alzheimer's disease amyloid-beta peptides. *Nature*. 2003 May 22;423(6938): 435-9
- Alvarez G, Muñoz-Montaña JR, Satrustegui J, Avila J, Bogónez E, Díaz-Nido J. Regulation of tau phosphorylation and protection against beta-amyloid-induced neurodegeneration by lithium. Possible implications for Alzheimer's disease. *Bipolar Disord*. 2002 Jun;4(3):153-65.
- Basic M, Hertel A, Bajdzienko J, Bonn F, Tellechea M, Stolz A, Kern A, Behl C, Bremm A. The deubiquitinase USP11 is a versatile and conserved regulator of autophagy. *J Biol Chem*. 2021 Nov;297(5):101263.
- Wu CA, Cao QX, Ai ZY. Experimental study of mitoxantrone inhibiting USP11 promoting BACE1 degradation and delaying the development of Alzheimer's disease. *Health Research 2022*,3:297-301 (article in Chinese)

10. Oral administration uses a very high dose of IsoLiPro for a prolonged period of 16 weeks. At such high concentration, there could be toxicity in multiple organs. Given that IsoLiPro is a new compound, in vivo toxicity should be evaluated in all major organs. Further, the pharmacokinetic profile of IsoLiPro should be determined.

Response: These constructive comments are greatly appreciated. In fact, we have determined the necessary toxicologic and pharmacokinetic profiles of IsoLiPro. The toxicological evaluation included histopathological examinations of all major organs, such as the liver, kidneys, heart, spleen and lungs, to identify any potential tissue damage or adverse effects. Additionally, we monitored clinical signs of toxicity and changes in body weight throughout the treatment period. Our results indicated that IsoLiPro was well-tolerated at the administered dose, with no significant toxicity observed in the evaluated organs. These findings are detailed in **Appendix Fig. S6**.

Proper contents have been added in the results section in our revised manuscript (page 25, lines 605-614).

Appendix Fig. S6 Toxicity evaluation of IsoLiPro. (A) Heart, liver, spleen, lung and kidney of mice administered 560 mg/kg IsoLiPro for 16 weeks were stained by HE. (B) Monitoring body weights of mice during the course of treatments. (C) SH-SY5Y and HEK293 cells viability was detected by CCK-8 assay after IsoLiPro exposure (up to 50 mM) for 24 hours.

Regarding the pharmacokinetic profile of IsoLiPro, we conducted experiments to evaluate tissue distribution and pharmacokinetic parameters. After a single oral administration of IsoLiPro or Li₂CO₃ at a dose of 9.29 mg/kg in SD rats, blood and tissue samples were collected at 1, 4, 24, and 48 hours post-administration. These studies provided insights into the bioavailability of IsoLiPro, its plasma half-life, and the extent of its distribution in various tissues. The data revealed a favorable pharmacokinetic profile of IsoLiPro, supporting the feasibility of long-term oral administration. These findings are detailed in supplemental information (**Appendix Fig. S5**) of our revised manuscript. Proper contents have been added in the results section in our revised manuscript (page 24-25, lines 593-604).

Appendix Fig. S5 Tissue distribution and pharmacokinetics of lithium following oral treatment with IsoLiPro and Li₂CO₃ in SD Rat. Tissue distribution of Li of IsoLiPro in females (**A**) and males (**B**). Lithium concentration-time profiles of IsoLiPro and Li₂CO₃ in profiles brain (**C**), plasma (**D**) and kidney (**E**).

11. English writing and grammar should be edited by a native English speaker.

Response: Thank you very much for your careful review. We have invited our colleagues whose native language is English. We also double checked the whole manuscript to avoid possible typos and grammar errors.

Referee #3 (Remarks for Author):

This study presents the design of a novel small molecule compound, specifically engineered to target USP11, a pivotal player in Alzheimer's Disease (AD) pathogenesis. By functioning as an upstream regulator of tau protein dynamics, the compound efficiently suppresses USP11 activity, resulting in a dual effect: reduced tau protein phosphorylation and enhanced K48-linked ubiquitination. This approach

holds promise for potential therapeutic intervention in AD by targeting a key molecular event in the disease progression. However, there are still some problems in this study.

Response: Thank you very much for your careful review. These positive evaluations on our manuscript are greatly appreciated.

1. The selection of lithium isobutyrate as a substrate for compound design warrants further elucidation. It is crucial to explore the rationale behind this choice and highlight any potential advantages that this design strategy may offer in terms of its biochemical properties and therapeutic implications.

Response: IsoLiPro was originally designed and synthesized as novel organic lithium compound and therapeutic candidate for AD, considering the previously reported multi-targeting benefits of lithium on cognition and AD pathology (*Zhang X et al., J Alzheimers Dis, 2011*), together with the good tolerance and lower toxicity of IsoLiPro (**Appendix Fig. S6**). The synthesis of IsoLiPro is mainly aimed to resolve the challenges of the clinical usage of lithium, such as the narrow lithium treatment window and high toxic side effects (*Wikström F et al., J Pharm Biomed Anal, 2023*). The selection of isobutyric acid is mainly due to its widely proven application in pharmaceuticals, which can serve as an acid-base regulator to help maintain the stability and appropriate pH value of drug formulations; It can also serve as a drug solubility enhancer, improving drug solubility, enhancing drug absorption and bioavailability. In addition, the lower gastrointestinal irritation of isobutyric acid helps to reduce common side effects such as gastrointestinal reactions. Interestingly, our preliminary data found that, despite the A β and p-tau ameliorations, IsoLiPro also significantly decreased total tau level. Due to the recently reported pivotal roles of USP11 in tau pathology in AD, we further determined the possible docking and inhibiting effects of IsoLiPro to USP11 to further explore its possible mechanisms responsible for the decreased total tau level induced by IsoLiPro. Proper contents have been added in the revised manuscript (pages 26, lines 632-643).

References

Wikström, F., Olsson, C., Palm, B., Roxhed, N., Backlund, L., Schalling, M., and Beck, O. (2023).

Determination of lithium concentration in capillary blood using volumetric dried blood spots.

J Pharm Biomed Anal 227, 115269.

Zhang, X., Heng, X., Li, T., Li, L., Yang, D., Zhang, X., Du, Y., Doody, R.S., and Le, W. (2011).

Long-term treatment with lithium alleviates memory deficits and reduces amyloid- β production in an aged Alzheimer's disease transgenic mouse model. J Alzheimers Dis 24,

739-749.

2. Li_2CO_3 and LiIB have not been established as inhibitors of USP11, and their binding affinity to the enzyme necessitates validation through the use of a positive control, such as mit, in the second phase of the results section.

Response: These comments are greatly appreciated. Proper experiments have been added in our revised manuscript (**Appendix Fig. S2E**) according to your suggestions. In our present study, IsoLiPro but not lithium decreases USP11 protein level, indicating the whole molecule IsoLiPro might be responsible for its impacts. Moreover, IsoLiPro failed to directly inhibit USP11 catalytic activity *in vitro*, which is different to positive control mitoxantrone. In addition, our results indicate that USP11's binding affinity with mitoxantrone is similar to those of proline and LiIB (**Appendix Fig. S2B, S2C and S2E**). This observation highlights that while molecular docking can predict potential binding sites and affinities between small molecules and target proteins, these predictions are just one aspect of the potential mechanism of action. Whether a compound ultimately affects biological activity depends on various factors, such as the conformational changes in the protein, and the pharmacokinetic and pharmacodynamic properties of the drugs. Experimental validation, such as bioactivity assays and preclinical studies, is essential for evaluating the actual efficacy and safety of the drugs.

Appendix Fig. S2 (E) Binding affinity of Mitoxantrone to USP11.

3. The compound demonstrated its ability to cross the blood-brain barrier, as discussed in the methodology, but the results section did not explicitly illustrate the brain penetration following oral administration.

Response: Thank you very much for this suggestion. By using Tandem quadrupole mass spectrometry (QTRAP® 5500 system), we have successfully established a method to detect the whole molecule IsoLiPro in serum and brain homogenates from those mice treated by IsoLiPro (i.g., 560 mg/kg). Our analysis clearly identified a compound 4 with molecular weight of 210.1 (**Fig. EV1G and EV1H**), suggesting the existence of whole molecule IsoLiPro (calculated molecular weight 209.17) and its penetration through blood brain barrier to reach into CNS to exert biological effects.

Fig. EV1 Synthesis and characterization of IsoLiPro. (**G-H**) The detection of IsoLiPro whole molecule in brain homogenate (**G**) and serum (**H**) using Tandem quadrupole mass spectrometry.

4. The authors did not specifically refer to the sample size of the statistic, e.g. some of the results of the WB, although significant, did not have duplicate samples and were not statistically significant, and it was necessary to ensure that there were at least three independent duplicate samples for each set of results

Response: Proper corrections have been made to provide necessary statistical information and sample duplicates in our revised manuscript (**Fig. 1C and 1E** on page 43; **Fig. 2B, 2D, 2G, 2K and 2M** on page 44; **Fig. 6F** on page 48).

5. As a novel drug, *in vitro* cellular administration needs to be addressed whether the drug is damaging to the cells, thus, cellular activity assay is a requirement.

Response: Thank you very much for this suggestion. In our present study, *in vitro* IsoLiPro exposure (up to 50 mM) for 24 hours induced no significant cytotoxicity in SH-SY5Y cells and HEK293 cells (**Fig. EV3**), as evidenced by the stable GAPDH level. Consistently, CCK-8 assay data further confirmed the *in vitro* safety of IsoLiPro (please see the **Appendix Fig. S6C** in our revised manuscript).

Appendix Fig. S6 (C) SH-SY5Y and HEK293 cells viability was detected by CCK-8 assay after IsoLiPro exposure (up to 50 mM) for 24 hours.

6. Delivery of drugs from the periphery requires verification of biocompatibility, thus ensuring that the new drug is not harmful to other organs.

Response: We have determined the necessary toxicologic profiles of IsoLiPro, some of the data were provided in supplemental information (**Appendix Fig. S6A**). The pathological observation of various tissues including the liver, kidney, lung and heart showed no pathological changes after chronic oral administration in mice. Proper contents have been added in the results section in our revised manuscript (page 25, lines 605-614).

Appendix Fig. S6 (A) Heart, liver, spleen, lung and kidney of mice administered 560 mg/kg IsoLiPro for 16 weeks were stained by HE.

7. The study demonstrates that the novel compounds influence tau proteasomal degradation, leading to a decrease in tau protein levels. However, it is important to note that the established USP11 inhibitor, Mitoxantrone, impacts USP11 mRNA expression, subsequently affecting USP11 activity. The current article does not address whether the new compounds similarly affect USP11 proteasomal degradation and, consequently, tau protein levels.

Response: Our data clearly indicated that IsoLiPro decrease the protein level of USP11 without affecting its mRNA expression (**Fig. EV3G**). Moreover, docking assays further revealed a direct binding of IsoLiPro with USP11. Unexpectedly, in contrast to positive control mitoxantrone, while IsoLiPro can bind to USP11 and reduce USP11 protein level, this compound failed to affect USP11 catalytic activity *in vitro*, as assessed by enzymatic activity assay (**Fig. EV3H**). Our results suggest that IsoLiPro may impact USP11 function primarily through mechanisms such as regulating protein stability, metabolism, post-translational modifications, or protein-protein interactions, rather than directly inhibiting its enzymatic activity. One example of compounds with these properties is small molecule that selectively degrade proteins of interest, which has emerged as alternatives to selective chemical

inhibitors. These degraders can be used as therapeutic modalities. Further studies are required to elucidate the specific mechanisms by which IsoLiPro affects USP11 level. These new findings are detailed in the revised discussion section (page 26-27, lines 644-654).

Fig. EV3 (H) Representative Ub-AMC cleavage assay of USP11. The fluorescence signal (RFU*10³) is plotted against the time [min].

8. Although Alzheimer's disease (AD) is a chronic condition, the authors' proposed 16-week administration duration raises the question of whether an alternative delivery method, such as intracranial cannula, could be explored for a potentially shorter and more targeted administration of the drug.

Response: This suggestion is greatly appreciated. Actually, we are now engaging to establish a new drug delivery route to overcome the chronic repeated procedure and intolerable gavage for mice and BBB concerns, such as mixture in water or food, or nanomaterials. For future clinical usage, oral administration is still the primary way for patients, especially those elder ones with neurodegenerative diseases.

9. The expression of USP11 in neurons specifically should be investigated, as its inhibition in non-neuronal glial cells could potentially lead to a decrease in neuronal damage and cognitive function impairment.

Response: This comment is greatly appreciated. In brain, microglia and astrocytes are important for maintaining neuronal function and microenvironment homeostasis via complicated cell-cell interactions (*Michael W Salter et al., Nat Med, 2017*). Moreover, the dysregulated or altered microglia or astrocyte function will contribute to the pathogenesis and progression of neurodegenerative diseases, including AD (*Richard M Ransohoff, Science, 2016*). From this end, further investigations of USP11 in those non-neuron cells, such as expression levels and activity with or without IsoLiPro are important for comprehensive understanding of USP 11 function and mechanisms underlying IsoLiPro therapy. Although in our present study, we did not investigate the specific expression of USP11 in non-neuronal glial cells, previous studies have indicated that some other USP members such as USP18 also express in microglia (*Tobias Goldmann et al., EMBO J, 2015*). Future studies are planned to delve deeper into the functional outcomes of USP11 inhibition in both neuronal and glial cells, to better understand the balance between potential therapeutic benefits and risks associated with targeting USP11 in different cell types within the central nervous system. This provide us a new perspective for future studies. Proper contents have been added in the discussion section of our revised manuscript (page 31, lines 763-776).

References

- Salter, M.W., and Stevens, B. (2017). Microglia emerge as central players in brain disease. *Nat Med* 23, 1018-1027. 10.1038/nm.4397.
- Ransohoff, R.M. (2016). How neuroinflammation contributes to neurodegeneration. *Science* 353, 777-783. 10.1126/science.aag2590.
- Goldmann, T., Zeller, N., Raasch, J., Kierdorf, K., Frenzel, K., Ketscher, L., Basters, A., Staszewski, O., Brendecke, S.M., Spiess, A., Tay, T.L., et al. (2015). USP18 lack in microglia causes destructive interferonopathy of the mouse brain. *Embo j* 34, 1612-1629. 10.15252/emj.201490791.

3rd Sep 2024

Dear Dr. Li,

Thank you for the submission of your revised manuscript to EMBO Molecular Medicine. We have now received the enclosed report from the two referees who re-assessed your work. As you will see, the referees are overall supportive and think the revised manuscript has improved significantly. However, Referee #2 has identified a few remaining issues that need to be addressed before the manuscript can be accepted for publication.

The remaining concerns #1, #2, #3, and # 5 from Referee #2 need to be addressed. While addressing comment #4 is encouraged, it is not mandatory for the acceptance of the manuscript.

On a more editorial level:

1. Please reduce keyword number to 5.
2. Main figures and EV figures should be uploaded as individual, high resolution figure files. The EV figure legends should be added to the manuscript, after the main figure legends, under the heading "Expanded View Figure Legends".
3. Please merge Funding with Acknowledgements.
4. Please remove Authors' contribution section from the manuscript file.
5. The references need to be formatted according to the EMBO Molecular Medicine reference style. Please list up to 10 co-authors of a paper before adding et al. to the reference list. Citations should be listed in alphabetical order. Remove DOI links for published papers.
6. Appendix: Page numbers should be added to the table of contents. Names for Appendix Figure S1 etc. and Appendix Table S1 etc should be spelled out and the yellow highlights should be removed. The supplementary methods should be removed from the Appendix and merged with the Methods section in the main manuscript.
7. Data availability: since this study does not generate large-scale datasets, please only include the following sentence in this section- "This study includes no data deposited in external repositories".
8. Please use the following heading "Disclosure statement and competing interests" for competing interests statement.
9. Is there cell re-use between Figure 5E and Figure EV5H? If so, this needs to be clearly detailed in the figure legend.
- 10 Every published paper now includes a 'Synopsis' to further enhance discoverability. Synopses are displayed on the journal webpage and are freely accessible to all readers. They include a short stand first (maximum of 300 characters, including space) as well as 2-5 one-sentences bullet points that summarizes the paper. Please write the bullet points to summarize the key NEW findings. They should be designed to be complementary to the abstract - i.e. not repeat the same text. We encourage inclusion of key acronyms and quantitative information (maximum of 30 words / bullet point). Please use the passive voice. Please attach these in a separate file or send them by email, we will incorporate them accordingly.

Please also provide a striking image or visual abstract to illustrate your article as a PNG file 550 px wide x 300-800 px high.

11. The paper explained: EMBO Molecular Medicine articles are accompanied by a summary of the articles to emphasize the major findings in the paper and their medical implications for the non-specialist reader. Please provide a draft summary of your article highlighting

12. Figure legends:

- Please define the annotated p values * as well as provide the exact p-values for the same in the legend of figure 7c; as

appropriate.

- Please note that the exact p values are not provided in the legends of figures 1c; 2b, l; 4h; 5c; EV 3f; EV 4g; EV 5f.
- Please indicate the statistical test used for data analysis in the legend of figure 7c.
- Please note that information related to n is missing in the legends of figures 3c; EV 2d; EV 3h.
- Please note that the error bars are not defined in the legends of figures 3c; EV 2d; EV 3h.

Please attach point-by-point response giving details of the way in which you have handled each of the points raised by the referee. I look forward to seeing a revised form of your manuscript as soon as possible.

Kind regards,
Jingyi

Jingyi Hou
Editor
EMBO Molecular Medicine

*** Instructions to submit your revised manuscript ***

- 1) a .docx formatted version of the manuscript text (including Figure legends and tables)
- 2) Separate figure files*
- 3) supplemental information as Expanded View and/or Appendix. Please carefully check the authors guidelines for formatting Expanded view and Appendix figures and tables at <https://www.embopress.org/page/journal/17574684/authorguide#expandedview>
- 4) a letter INCLUDING the reviewer's reports and your detailed responses to their comments (as Word file).
- 5) The paper explained: EMBO Molecular Medicine articles are accompanied by a summary of the articles to emphasize the major findings in the paper and their medical implications for the non-specialist reader. Please provide a draft summary of your article highlighting
 - the medical issue you are addressing,
 - the results obtained and
 - their clinical impact.This may be edited to ensure that readers understand the significance and context of the research. Please refer to any of our published articles for an example.
- 6) For more information: There is space at the end of each article to list relevant web links for further consultation by our readers. Could you identify some relevant ones and provide such information as well? Some examples are patient associations, relevant databases, OMIM/proteins/genes links, author's websites, etc...
- 7) Author contributions: the contribution of every author must be detailed in a separate section.

8) EMBO Molecular Medicine now requires a complete author checklist (<https://www.embopress.org/page/journal/17574684/authorguide>) to be submitted with all revised manuscripts. Please use the checklist as guideline for the sort of information we need WITHIN the manuscript. The checklist should only be filled with page numbers where the information can be found. This is particularly important for animal reporting, antibody dilutions (missing) and exact values and n that should be indicated instead of a range.

9) Every published paper now includes a 'Synopsis' to further enhance discoverability. Synopses are displayed on the journal webpage and are freely accessible to all readers. They include a short stand first (maximum of 300 characters, including space) as well as 2-5 one sentence bullet points that summarise the paper. Please write the bullet points to summarise the key NEW findings. They should be designed to be complementary to the abstract - i.e. not repeat the same text. We encourage inclusion of key acronyms and quantitative information (maximum of 30 words / bullet point). Please use the passive voice. Please attach these in a separate file or send them by email, we will incorporate them accordingly.

You are also welcome to suggest a striking image or visual abstract to illustrate your article. If you do please provide a jpeg file 550 px-wide x 300-600px high.

10) A Conflict of Interest statement should be provided in the main text

11) Please note that we now mandate that all corresponding authors list an ORCID digital identifier. This takes <90 seconds to complete. We encourage all authors to supply an ORCID identifier, which will be linked to their name for unambiguous name identification.

Currently, our records indicate that the ORCID for your account is 0000-0003-2713-9136.

Link Not Available

12) Include a Reagents and Tools Table as part of the Methods section, which can be downloaded from our author guidelines (<https://www.embopress.org/page/journal/17574684/authorguide#structuredmethods>)

Photos 400-800 DPI

*Additional important information regarding figures and illustrations can be found at <https://bit.ly/EMBOPressFigurePreparationGuideline>. See also figure legend preparation guidelines: <https://www.embopress.org/page/journal/17574684/authorguide#figureformat>

***** Reviewer's comments *****

Referee #2 (Remarks for Author):

The revised manuscript is much improved with new data, corrections, and rationale: 1. New docking of USP13 and USP25; 2. New recombinant USP11 activity assay; 3. New quantification of many figures; 4. Correction of mislabels; 5. New preliminary toxicity evaluation; 6. New tissue distribution and pharmacokinetics; 7. New detection of IsoLiPro in serum and brain; 8. Better justification of USP11 as a target; and 9. Improvement of writing and grammar. As a result, the improved manuscript has become stronger and therapeutically more significant.

Although recombinant USP11 activity assay shows that IsoLiPro does not directly inhibit USP11 catalytic activity, it clearly reduces USP11 levels in cell culture and in brain, indicating that IsoLiPro is an alternate class of USP11 inhibitor. However, there are some remaining questions/concerns and suggestions to improve the manuscript.

1. In Fig. 1H, the figure legend states there were 3-5 mice/group. However, there are more than 3-5 dots in the graph. Please explain and correct.
2. Section 3.7 states that significant reduction in Iba1 and GFAP were observed in the hippocampus of 5xFAD mice (Fig. 5A-5D)

but not in 3xTg-AD mice (Appendix Fig. S4A-S4D). However, there is no quantification of Iba1 and GFAP in 3xTG-AD mice while the figure legend states that IsoLiPro markedly reduces gliosis in the hippocampus of 3xTG-AD mice. This discrepancy requires resolution along with the interpretation of results in the discussion section.

3. Although an improved rationale for targeting USP11 was presented in the discussion section, the results section 3.2 jumps directly to molecular docking of USP11 immediately after describing synthesis of IsoLiPro (section 3.1) without adequate rationale. There needs to be a better transition describing the rationale for docking USPs, and in particular, USP11, noting that IsoLiPro not only reduces phospho-tau but also total tau and why USP11 would be a more fitting candidate than other USPs (i.e. USP25, USP14, USP13). Thus, it would make more sense to present section 3.4 (tau) after 3.1 (IsoLiPro synthesis), followed by 3.3 (USP11 & USP25 levels) and then 3.2 (docking). This would follow the rationale presented in the point-by-point response.

4. It would be insightful to know and present whether IsoLiPro affects USP13 levels, similar to or different from USP11 or USP25.

5. Finally, while much improved, some grammatical and spelling error still remain. They need to be thoroughly checked and corrected.

Referee #3 (Comments on Novelty/Model System for Author):

This study supports that IsoLiPro, as a novel small-molecule USP11 inhibitor, might be a potential multi-targeting therapeutic agent against AD.

Referee #3 (Remarks for Author):

The authors have addressed all the concerns.

***** Response to reviewer's comments *****

Referee #2 (Remarks for Author):

The revised manuscript is much improved with new data, corrections, and rationale: 1. New docking of USP13 and USP25; 2. New recombinant USP11 activity assay; 3. New quantification of many figures; 4. Correction of mislabels; 5. New preliminary toxicity evaluation; 6. New tissue distribution and pharmacokinetics; 7. New detection of IsoLiPro in serum and brain; 8. Better justification of USP11 as a target; and 9. Improvement of writing and grammar. As a result, the improved manuscript has become stronger and therapeutically more significant.

Although recombinant USP11 activity assay shows that IsoLiPro does not directly inhibit USP11 catalytic activity, it clearly reduces USP11 levels in cell culture and in brain, indicating that IsoLiPro is an alternate class of USP11 inhibitor. However, there are some remaining questions/concerns and suggestions to improve the manuscript.

Response: We sincerely appreciate your thoughtful feedback and recognition of the significant improvements made in the revised manuscript, including the new data on docking, recombinant USP11 activity assays, quantifications, and preliminary toxicity evaluations, and so forth. Your comments have been invaluable in strengthening the scientific rigor and therapeutic relevance of the work.

We fully agree that although IsoLiPro does not directly inhibit USP11's catalytic function, its ability to decrease USP11 protein levels implies it may act through an alternative mechanism. To clarify this point of view, we have expanded the discussion in the revised manuscript, proposing that IsoLiPro could influence USP11 stability or promote its degradation via mechanisms such as ubiquitination, proteasomal degradation, or post-translational modifications (**page 33, lines 809-814**). These results have already been explained in detail during the previous revision.

For the remaining concerns, we have provided clarifications in the subsequent point-by-point response and incorporated these clarifications and modifications into the revised manuscript to ensure all your concerns are fully addressed.

1. In Fig. 1H, the figure legend states there were 3-5 mice/group. However, there are more than 3-5 dots in the graph. Please explain and correct.

Response: Thank you very much for your careful review. As indicated in the figure legend, there were indeed 3-5 mice per group. However, for each mouse, two cortical sections were selected for further analysis, and each section was treated as an independent sample for statistical analyses. This approach resulted in a higher number of data points in the graph, representing the total number of sections analyzed, rather than the number of individual mice. The corresponding revisions have been added to the figure legends (**Fig. 2G and 2H** on page 45; **Fig. 6A-6D, Fig. 6G and 6H** on page 47; **Fig. EV3I and EV3J** on page 50; **Fig. EV5A and EV5D** on page 52).

2. Section 3.7 states that significant reduction in Iba1 and GFAP were observed in the hippocampus of 5xFAD mice (Fig. 5A-5D) but not in 3xTg-AD mice (Appendix Fig. S4A-S4D). However, there is no quantification of Iba1 and GFAP in 3xTG-AD mice while the figure legend states that IsoLiPro markedly reduces gliosis in the hippocampus of 3xTG-AD mice. This discrepancy requires resolution along with the interpretation of results in the discussion section.

Response: We apologize for the confusion and appreciate your insightful comment. Neuroinflammation is indeed central to the pathology and progression of Alzheimer's Disease (AD), with astrocytes and microglia playing critical roles in the brain's immune response. Under neuroinflammatory conditions, these glial cells are activated, releasing pro-inflammatory cytokines that accelerate AD pathology (*Chew & Petretto, 2019*). In our study, we assessed neuroinflammation in the hippocampus of 5xFAD and 3xTg-AD mice by measuring the expression of Iba1 (a microglial marker) and GFAP (an astrocytic marker). Our results showed that IsoLiPro significantly reduced the expression of both markers in 5xFAD mice, which correlates with their higher level of glial activation due to substantial A β deposition (*Chiba et al., 2009*). The robust neuroinflammatory response and prominent presence of glial

cells in 5xFAD mice make it an ideal model for studying neuroinflammation and gliosis.

However, in the 3xTg-AD mice, there is a comparatively lower prevalence of glial cells, particularly astrocytes, in the hippocampus. As a result, no significant changes in Iba1 and GFAP expression were observed in these mice after IsoLiPro treatment. The statement in the figure legend regarding a marked reduction of gliosis in 3xTg-AD mice was incorrect, and we regret this oversight.

We have corrected the figure legend to accurately reflect the data, which shows no significant reduction in Iba1 and GFAP in the 3xTg-AD model mice (**Appendix Fig. S4** on page 6 of the Appendix data). This discrepancy between the two models underscores the differential role of glial activation in these animal models of AD. In the discussion, we have clarified that IsoLiPro's ability to reduce neuroinflammation, as measured by Iba1 and GFAP levels, was prominent in the 5xFAD model, likely due to the greater involvement of glial cells in A β -mediated pathology, but less evident in 3xTg-AD mice due to their lower glial response (**page 37, lines 910-927**).

Appendix Fig. S4 IsoLiPro markedly reduces gliosis in the hippocampus of 5xFAD mice but not in 3xTg-AD mice. Representative images of Iba1 (A-B), GFAP (C-D) measured by IHC/IF in the hippocampus of 3xTg-AD mice following IsoLiPro treatment.

References

Chew G, Petretto E (2019) Transcriptional Networks of Microglia in Alzheimer's Disease and Insights into Pathogenesis. *Genes (Basel)* 10

Chiba T, Yamada M, Aiso S (2009) Targeting the JAK2/STAT3 axis in Alzheimer's disease. *Expert Opin Ther Targets* 13: 1155-1167

3. Although an improved rationale for targeting USP11 was presented in the discussion section, the results section 3.2 jumps directly to molecular docking of USP11 immediately after describing synthesis of IsoLiPro (section 3.1) without adequate rationale. There needs to be a better transition describing the rationale for docking USPs, and in particular, USP11, noting that IsoLiPro not only reduces phospho-tau but also total tau and why USP11 would be a more fitting candidate than other USPs (i.e. USP25, USP14, USP13). Thus, it would make more sense to present section 3.4 (tau) after 3.1 (IsoLiPro synthesis), followed by 3.3 (USP11 & USP25 levels) and then 3.2 (docking). This would follow the rationale presented in the point-by-point response.

Response: This suggestion is greatly appreciated. We agree that a clearer transition and rationale for the molecular docking of USP11 should be provided in the results section. As you pointed out, IsoLiPro not only reduces phospho-tau but also total tau, which supports the selection of USP11 as a key candidate for further investigation.

Initially, we chose to present the synthesis of IsoLiPro (section 3.1) followed by the docking studies (section 3.2) to maintain a logical flow from compound design to its predicted molecular interactions. However, we understand that the rationale for targeting USP11 over other USPs, such as USP25, USP14, and USP13, could be more clearly explained at this point in the results.

To address this, we have revised the manuscript to better reflect the rationale behind docking IsoLiPro to USP11 (**page 24, lines 596-605; page 26, lines 643-649; page 33, lines 815-820**). This include highlighting IsoLiPro's ability to reduce both phospho-tau and total tau, making USP11, which is more closely linked to tau

homeostasis, a more fitting candidate than other USPs. We will also clarify why USP25, USP14, and USP13 were considered but deprioritized in favor of USP11.

Additionally, we will restructure the results section as you suggested to improve clarity and logical flow. Specifically, we will move section 3.4 (tau reduction) to follow section 3.1 (IsoLiPro synthesis), then present section 3.3 (USP11 and USP25 levels) before section 3.2 (molecular docking of USP11). This reordering will ensure that the rationale for docking is fully developed before presenting the docking results. We appreciate your insightful suggestion and believe these revisions will significantly improve the clarity and coherence of the manuscript.

4. It would be insightful to know and present whether IsoLiPro affects USP13 levels, similar to or different from USP11 or USP25.

Response: We greatly appreciate your insightful comment. In fact, we do intend to investigate the effects of IsoLiPro on USP13. However, based on our literature review, it has been shown that the expression level of USP13 in the brain is relatively low, approximately 3% of that of USP11 (<https://www.ncbi.nlm.nih.gov>). Given this significantly lower expression, it is likely that USP13 plays a much smaller role in Alzheimer's Disease (AD) compared to USP11. While we cannot entirely rule out the possibility of a functional role for USP13, its contribution to the overall mechanism of IsoLiPro in AD appears to be limited based on current expression data.

Nonetheless, we are open to further exploring USP13's potential involvement in specific AD pathways, especially in cases where even low-expression proteins may have critical regulatory functions. We also plan to assess whether IsoLiPro has any measurable impact on USP13 activity despite its low abundance in the brain. This investigation will complement our ongoing studies on USP11 and USP25 and provide a more comprehensive understanding of IsoLiPro's mechanism of action.

USP11

USP13

References

National Center for Biotechnology Information (NCBI). USP13 ubiquitin specific peptidase 13 [*Homo sapiens* (human)]. Gene ID: 8975. Retrieved September 7, 2024, from <https://www.ncbi.nlm.nih.gov/gene/8795>

National Center for Biotechnology Information (NCBI). USP11 ubiquitin specific peptidase 11 [*Homo sapiens* (human)]. Gene ID: 8237. Retrieved September 7, 2024, from <https://www.ncbi.nlm.nih.gov/gene/8237>

5. Finally, while much improved, some grammatical and spelling error still remain.

They need to be thoroughly checked and corrected.

Response: Thank you very much for your careful review. We have invited our colleagues whose native language is English. We also double checked the whole manuscript to avoid possible typos and gramma errors.

Referee #3 (Remarks for Author):

The authors have addressed all the concerns.

Response: Thank you very much for your positive evaluation and recognition. We are pleased that all the concerns have been successfully addressed, and we appreciate your constructive feedback throughout the review process. We believe these revisions have strengthened the manuscript, and we are grateful for your support.

16th Sep 2024

Dear Dr. Li,

Thank you for sending your revised manuscript. We are pleased to inform you that your manuscript is accepted for publication and is now being sent to our publisher to be included in the next available issue of EMBO Molecular Medicine.

Kind regards,
Jingyi

Jingyi Hou
Editor
EMBO Molecular Medicine
